# DDK/Hsk1 phosphorylates and targets fission yeast histone deacetylase Hst4 for degradation to stabilize stalled DNA replication forks

**Shalini Aricthota[1,2], Devyani Haldar[1]\***

[1]Laboratory of Chromatin Biology and Epigenetics, Centre for DNA Fingerprinting and Diagnostics, Hyderabad, India; [2]Graduate Studies, Manipal Academy of Higher Education, Manipal, India

**Abstract** In eukaryotes, paused replication forks are prone to collapse, which leads to genomic instability, a hallmark of cancer. Dbf4-dependent kinase (DDK)/Hsk1[Cdc7] is a conserved replication initiator kinase with conflicting roles in replication stress response. Here, we show that fission yeast DDK/Hsk1 phosphorylates sirtuin, Hst4 upon replication stress at C-terminal serine residues. Phosphorylation of Hst4 by DDK marks it for degradation via the ubiquitin ligase SCF[pof3]. Phosphorylation-defective *hst4* mutant (*4SA-hst4*) displays defective recovery from replication stress, faulty fork restart, slow S-phase progression and decreased viability. The highly conserved fork protection complex (FPC) stabilizes stalled replication forks. We found that the recruitment of FPC components, Swi1 and Mcl1 to the chromatin is compromised in the *4SA-hst4* mutant, although whole cell levels increased. These defects are dependent upon H3K56ac and independent of intra S-phase checkpoint activation. Finally, we show conservation of H3K56ac-dependent regulation of Timeless, Tipin, and And-1 in human cells. We propose that degradation of Hst4 via DDK increases H3K56ac, changing the chromatin state in the vicinity of stalled forks facilitating recruitment and function of FPC. Overall, this study identified a crucial role of DDK and FPC in the regulation of replication stress response with implications in cancer therapeutics.

**\*For correspondence:**
devyani@cdfd.org.in

**Competing interest:** The authors declare that no competing interests exist.

## Introduction

DNA replication forks stall at various impediments including damaged DNA templates, DNA-protein complexes and secondary structures in chromatin generating replication stress (*Iyer and Rhind, 2013*; *Zeman and Cimprich, 2014*). In response to replication stress, the intra-S phase checkpoint is activated to maintain forks in a replication competent state and prevent fork collapse. The major checkpoint sensor ATR/Rad3 is active even in unperturbed S-phase to help recover from endogenous replication fork impediments. The progression of replication fork is dependent upon the chromatin state. However, in response to replication fork stalling, cross-talks of chromatin with replisome is elusive. DDK (Dbf4-dependent kinase) is a highly conserved serine/threonine kinase involved in the regulation of DNA replication through phosphorylation of MCM (mini chromosome maintenance) helicase (*Sheu and Stillman, 2006*; *Yabuuchi et al., 2006*). The role of DDK kinase Cdc7/Hsk1 is dependent upon its cell cycle regulated partner Dbf4/Dfp1 (*Jackson et al., 1993*). DDK-dependent replication mechanisms are well studied across all model systems. However, whether DDK has any positive role in the replication stress response pathway is still a debate. In *Saccharomyces cerevisiae (S. cerevisiae)*, reports show that Rad53 targets DDK to inhibit its kinase activity and limits late origin firing (*Jares et al., 2000*; *Kihara et al., 2000*; *Tsuji et al., 2008*; *Weinreich and Stillman, 1999*; *Zegerman and*

*Diffley, 2010*). Similarly, in *Schizosaccharomyces pombe (S. pombe)*, Cds1 phosphorylates Hsk1 upon HU treatment (*Matsumoto et al., 2010*; *Snaith et al., 2000*). However, the Rad53/Cds1 activation is reduced in the *cdc7/hsk1-89* mutant (*Yamada et al., 2014*). On the contrary, studies in humans show that DDK complex formation, chromatin association, and kinase activity are not perturbed after HU treatment (*Lee et al., 2012*; *Tenca et al., 2007*; *Tsuji et al., 2008*; *Yamada et al., 2013*). It has been reported that DDK helps initiate checkpoint signaling by aiding ssDNA formation (*Sasi et al., 2018*). Recently, DDK inhibition has been shown to be detrimental for human cells in S-phase and its role in fork remodeling during replication stress has been established (*Jones et al., 2021*).

The fork protection complex (FPC) consists of three members Timeless (Tim)/Tipin/Claspin in human, Tof1/Csm3/Mrc1 in *S. cerevisiae* while Swi1/Swi3/Mrc1 in *S. pombe* (*Bastia et al., 2016*; *Leman and Noguchi, 2012*; *Noguchi et al., 2004*). The function of FPC is critical under conditions of fork stress and also during normal, unperturbed cell cycle (*Lou et al., 2008*; *Tourrière et al., 2005*). Another replisome factor, And-1/Ctf4/Mcl1 is also a part of FPC as it functions as pol alpha accessory factor (*Gosnell and Christensen, 2011*; *Tanaka et al., 2009*). It has been reported that the Tim and Claspin are overexpressed in cancers and help in adaptability under replication stress (*Bianco et al., 2019*). The mechanisms regulating FPC in tumor cells are unknown. In *S. pombe*, deletion of *hsk1*, *swi1*, and *swi3* leads to decreased Cds1 activation. Hsk1 physically interacts with Swi1 and Swi3, however, it remains unclear how these proteins regulate replication stress response molecularly and whether they have functions independent of checkpoint (*Dolan et al., 2010*; *Matsumoto et al., 2005*). It has also been shown that in the absence of FPC, there is a coordinated degradation of replisome components via proteasome (*Roseaulin et al., 2013b*). Protein degradation plays a pivotal role in the regulation of various cellular processes (*Hershko et al., 2000*). The ubiquitin-proteasome system consists of a ubiquitin-activating enzyme (E1), ubiquitin-conjugating enzyme (E2), and ubiquitin ligase (E3) enzyme which polyubiquitinates the substrate proteins and marks them for degradation by the 26 S proteasome. The E3 ubiquitin ligases recognize specific substrate proteins and target these for proteolysis. The ubiquitin ligase, SCF (Skp1-Cdc53/Cullin-1-F-box) which recognizes phosphorylated substrates, regulates several S phase events during cell cycle progression (*Petroski and Deshaies, 2005*).

The role of chromatin modifications and their regulators in replication stress response is less studied (*Fournier et al., 2018*). The Sir2 family of conserved $NAD^+$-dependent protein deacetylases comprises of three members, Sir2, Hst2, and Hst4 in *S. pombe,* whereas it has seven members in mammals (SIRT1-7) called sirtuins. Deletion of *S. pombe hst4* causes S phase delay and sensitivity to replication stress causing agents (*Haldar and Kamakaka, 2008*; *Konada et al., 2018*). Hst4 levels are downregulated during unperturbed S phase and on MMS treatment (*Haldar and Kamakaka, 2008*; *Konada et al., 2018*). *S. pombe* Hst4 and Hst3/Hst4 (budding yeast) functions in preserving genome integrity by directly deacetylating H3K56 and thereby affecting chromatin state (*Maas et al., 2006*; *Masumoto et al., 2005*; *Miller et al., 2006*; *Xhemalce et al., 2007*){*Maas et al., 2006* #29; *Masumoto et al., 2005* #30; *Maas et al., 2006* #29}. Acetylation of H3K56 is a cell cycle regulated modification and is required for cell survival, replication stress response and nucleosome assembly (*Han et al., 2007b*; *Masumoto et al., 2005*; *Xu et al., 2005*). Acetylation of H3K56 is catalyzed by Rtt109, an acetyl transferase which requires a H3-H4 chaperone, Asf1 (*Sutton et al., 2003*) for this acetylation (*Driscoll et al., 2007*; *Han et al., 2007a*; *Han et al., 2007b*; *Recht et al., 2006*; *Tsubota et al., 2007*; *Xhemalce et al., 2007*). H3K56ac is required for completion of DNA replication and acts in the branch of S phase checkpoint (*Thaminy et al., 2007*). H3K56ac is known as the activator of transcription as it helps in turnover of promoter-proximal nucleosomes (*Topal et al., 2019*; *Xu et al., 2005*). The role of H3K56ac in mammals is less understood because it is not as abundant as it is in yeast (*Das et al., 2009*; *Vempati et al., 2010*; *Yuan et al., 2009*).

In this report, we identified a positive role of DDK in regulating replication stress response which is independent of its checkpoint functions. We show that in response to fork stalling, DDK/Hsk1 phosphorylates Hst4 at C-terminal serine residues leading to the formation of phosphodegron for recognition by SCF[pof3] E3 ubiquitin ligase and degradation via proteasome. We establish that degradation of Hst4 helps in stabilization of FPC components, Swi1 and Mcl1 upon replication stress via H3K56 acetylation at the chromatin. We found that Hst4 exhibit synthetic genetic interaction and physical interaction with FPC components, Swi1 and Swi3 in vitro. Hst4 may regulate the expression of FPC components in an H3K56 acetylation-dependent manner. We have also observed that these functions are independent of intra-S phase checkpoint. Further, we have shown conservation of regulation of

human FPC components via H3K56ac by knocking down Asf1 in U2OS cells. Taken together, our work revealed a potential role of H3K56ac in the regulation of FPC and established a positive role of DDK in regulating replication stress by modulating chromatin to stabilize DNA replication forks and thereby maintaining genome stability.

## Results

### DNA replication fork stalling cause downregulation of Hst4

The stability of DNA replication forks is maintained by an intricate interplay of the replisome, FPC and chromatin. We have previously shown that the level of sirtuin Hst4 decreases upon treatment with methylmethane sulfonate (MMS) which cause replication stress (*Haldar and Kamakaka, 2008*). To test whether Hst4 level is specifically controlled in response to MMS or the cellular regulation in response to replication fork stalling, asynchronously grown wild-type *S. pombe* cells were treated with MMS as well as another replicative stress causing agent, Hydroxyurea (HU) which depletes dNTP pools to stall DNA replication. Hst4 was downregulated upon HU treatment similar to MMS (*Figure 1A*). Further, we analyzed the kinetics of Hst4 downregulation upon MMS treatment and observed that it was downregulated after 90 min (*Figure 1B*). The kinetics of regulation of Hst4 on HU treatment was similar to MMS (*Figure 1C*). Since Hst4 is already known to be downregulated during the unperturbed S phase of the cell cycle, its level was examined on MMS treatment to check the degradation kinetics during the S phase in the presence of damage. Wild-type cells were synchronized at the G2 phase of the cell cycle and released in the S phase with and without MMS. Interestingly, the downregulation of Hst4 was accelerated by 30 min upon MMS treatment (*Figure 1D and E*), although MMS slowed S phase progression as shown by the corresponding flow cytometry profiles of cell cycle progression (*Figure 1F*). Unlike in untreated cells, where Hst4 can be observed at 60 min, on MMS treatment it is downregulated at this timepoint, however stable at 30 min (*Figure 1—figure supplement 1A, B*). The S-phase progression was further confirmed by performing septation index analysis of the corresponding timepoints as shown in *Figure 1—figure supplement 1C*. It has been previously reported that MMS-induced toxicity is dependent on replication and that MMS slows replication fork progression (*Cortez, 2005*; *Stokes et al., 2002*). To establish the dependence of regulation of Hst4 on DNA replication-induced stress, cells arrested at the G2 phase were treated with MMS and Hst4 level was examined. The Hst4 levels remained same in cells arrested at G2 phase upon MMS treatment while the asynchronous cells from the same genetic background showed downregulation of Hst4 upon MMS treatment (*Figure 1G*). The corresponding flow cytometry profile is being shown in *Figure 1H*. Overall, these results suggest that Hst4 level specifically decreased in response to DNA replication fork stalling during normal S phase as well as upon treatment with fork stalling agents.

### Proteasome-dependent degradation of Hst4 upon replication stress

Replication stress does not regulate Hst4 transcriptionally as the transcript level of the gene was not significantly altered upon MMS treatment (*Figure 2—figure supplement 1A*). Therefore, we hypothesized that replication stress targets this chromatin regulator for degradation via the proteasome. To investigate the stability of Hst4 in wild-type cells, the half-life of the protein was determined in the presence or absence of MMS using cycloheximide (CHX), which blocks new protein synthesis. Hst4 was degraded at 120 minutes in untreated wild-type cells, whereas the proteolysis was accelerated upon MMS treatment (*Figure 2A*), showing Hst4 was significantly degraded upon replicative stress. The corresponding quantitation from three independent experiments is shown in the *Figure 2—figure supplement 1B*. To examine the role of proteasome in the degradation of Hst4, *mts2-1* temperature sensitive cells (in which proteasome is inactive at non-permissive temperature) carrying genomic TAP-tagged Hst4 (TAP-Hst4) were grown at non-permissive temperature and western blot was performed (*Takayama et al., 2010*). Hst4 was significantly stabilized in *mts2-1* strain in the absence or presence of MMS (*Figure 2B and C*). We next investigated whether Hst4 is ubiquitinated in these cells. To pull down ubiquitinated proteins, these strains were transformed with pREP1-His6-ubiquitin or empty vector and Ni-NTA column was used to purify ubiquitinated proteins under denaturing conditions (*Takayama et al., 2010*). Hst4 was found to be ubiquitinated under normal as well as upon MMS treatment (*Figure 2D*). Immunoblotting with anti-his antibody is shown as an expression control. Overall, these results demonstrate that Hst4 is ubiquitinated in vivo and degraded via the proteasome

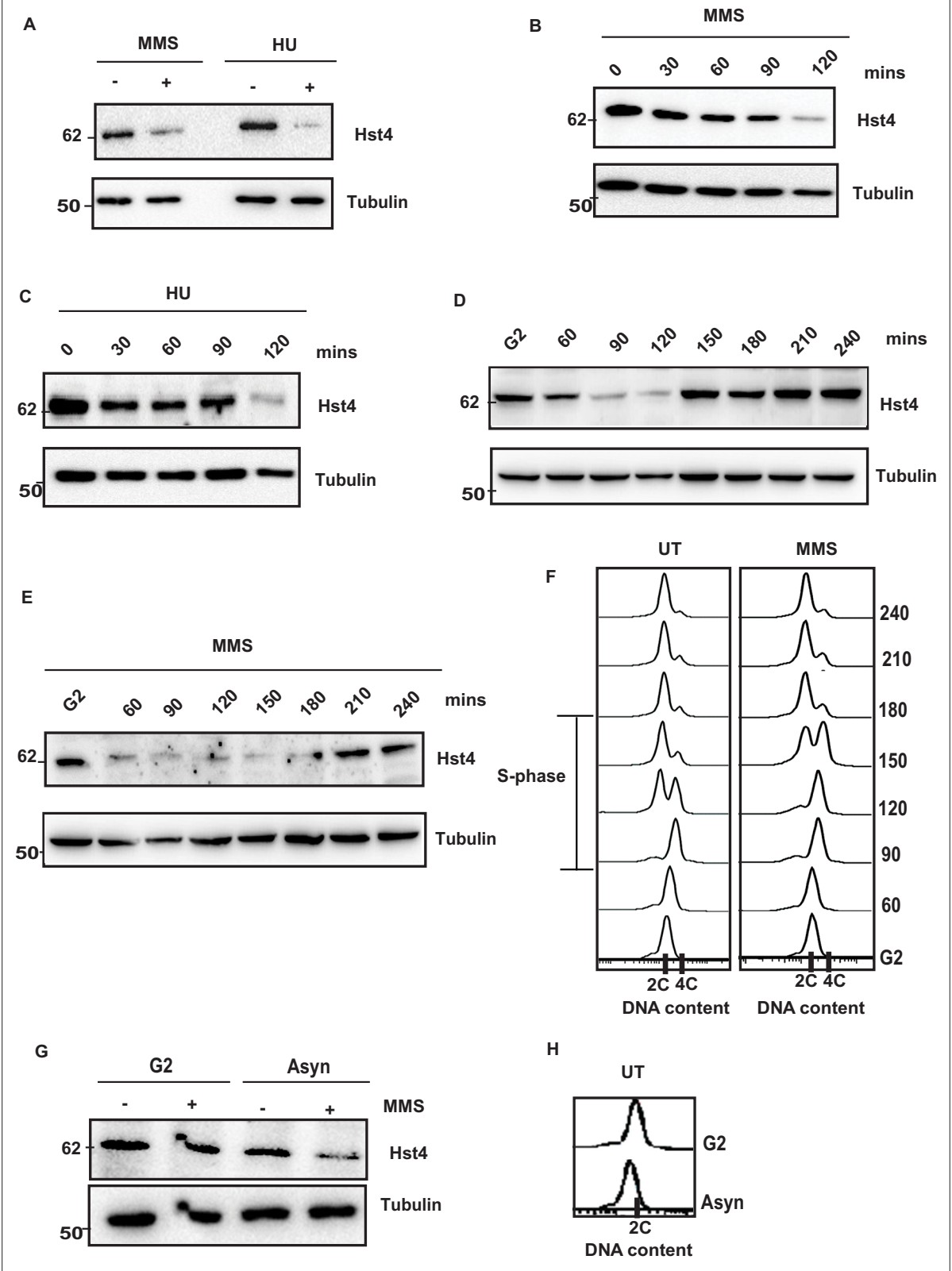

**Figure 1.** Replication stress leads to downregulation of Hst4. (**A**) Wild-type (wt) strain (ROP238) containing TAP-tagged Hst4 was grown to 0.5 O.D., treated with 0.015 % MMS and 10 mM HU for 2 hr, respectively, whole cell extracts were prepared and immunoblotted with anti-PAP antibody to detect TAP-tagged Hst4. (**B**) (**C**) wt strain (ROP238) containing TAP-tagged Hst4 was grown to 0.5 O.D., treated with 0.015 % MMS and 10 mM HU, extracts were prepared from cells collected at indicated time points and western blot was performed. (**D**, **E**) The *cdc25-22* cells expressing TAP-tagged Hst4 (DHP38)

*Figure 1 continued on next page*

*Figure 1 continued*

were first synchronized at G2 phase and then released into the S phase in the absence or presence of 0.015 % MMS, cells were collected at indicated time points and western blot was performed. (**F**) Flow cytometry profile of experiments in (**D and E**). (**G**) The *cdc25-22* cells (DHP38) were synchronized at G2 phase by growing cells at 36 °C, after 2 hr 0.015 % MMS was added and grown for another 2 hr, asynchronous cells grown at 25 degrees were used for this experiment. (**H**) Flow cytometry profile of experiment in (**G**).

The online version of this article includes the following figure supplement(s) for figure 1:

**Source data 1.** Uncropped western blot images for *Figure 1A– E and G*.

**Figure supplement 1.** S-phase dependent downregulation of Hst4 upon MMS treatment.

pathway. There are various mechanisms of proteasome dependent degradation of proteins. The mechanisms which operate in cell-cycle-dependent manner are APC and SCF-dependent E3 ubiquitin ligases (*Hershko et al., 2000*). Since, the regulation of Hst4 is dependent on S phase, we hypothesized that SCF E3 ubiquitin ligases could play a role in the phosphorylation dependent degradation of Hst4. Therefore, we performed bioinformatic analysis and observed potential phosphorylation sites at the C-terminus of Hst4. We made a truncated protein in which the C-terminal amino acids 353–415 are deleted. . The truncated protein was not degraded upon MMS treatment while its full-length counterpart was degraded, confirming the requirement of the C-terminal domain for degradtion of Hst4 (*Figure 2E and F*). We further deleted amino acids 398–415 (T1Δ-Hst4) and 371–415 (T2Δ-Hst4) and concluded that the minimum residues required for degradation are present between 352 and 370 amino acids as the truncations, T1 and T2 did not lead to complete stabilization of Hst4 (*Figure 2—figure supplement 1C,D*). We have also studied the subcellular localization of FL-Hst4 and CΔ-Hst4 and looked at the stability of these proteins upon MMS treatment by immunofluorescence. We observed that both the proteins are localizing perfectly to the nucleus, however, on MMS treatment, FL-Hst4 is degraded while CΔ-Hst4 is stabilized (*Figure 2G*, *Figure 2—figure supplement 1E*). Overall, these results showed that replication stress targeted Hst4 to proteasome-dependent degradation and the signal for degradation reside in the C terminus amino acids 352–370.

## DDK/Hsk1 phosphorylates Hst4 in response to replication stress at serine residues in the C terminus

The proteasome-dependent degradation machinery recognizes phosphorylated target proteins. We have performed potential phosphorylation motif search using Eukaryotic Linear Motif (ELM) software which is used to search functional sites in proteins. Bioinformatics analysis of the C-terminus of Hst4 revealed putative consensus phosphorylation sites of Hsk1 and Gsk3-Beta in the C-terminus of the protein, therefore, we checked whether Hst4 is phosphorylated upon MMS treatment and targeted for degradation. To test whether Hst4 is phosphorylated upon MMS treatment, TAP-Hst4 was immunoprecipitated from the cells grown in the absence and presence of MMS and the phosphorylation status was checked with phosphoserine antibody. We observed phosphorylation of Hst4 at 1 hr of MMS treatment (*Figure 3A*). We also subjected the immunoprecipitated Hst4 sample to lambda phosphatase treatment and observed appearance of a lower mobility form (*Figure 3B*). To check whether Hst4 is phosphorylated in the unperturbed S-phase of the cell cycle, we blocked wild-type cells at G2 phase, then released into cell cycle synchronously and collected cells at G2, 30, 60, and 90 min. As our results had (*Figure 1D*) indicated, Hst4 was degraded by 90 min. The TAP-Tagged Hst4 was immunoprecipitated and western blot was performed using phosphoserine antibody. Our results show that the phosphorylated band of Hst4 appears at 60 min (*Figure 3C*). Next, we wanted to identify the kinase which phosphorylates Hst4. Since, the C-terminal of Hst4 contained sites of Gsk3-Beta, we checked the level of Hst4 in *gsk3Δ* cells after MMS treatment. The degradation of Hst4 was not affected significantly in *gsk3Δ* cells (*Figure 3—figure supplement 1A*). Hsk1 is a major regulator of DNA replication initiation, it also mediates intra-S phase checkpoint response to DNA replication stress (*Matsumoto et al., 2005*; *Matsumoto et al., 2010*; *Snaith et al., 2000*). Since, Hst4 was degraded in a replication dependent manner and putative Hsk1 kinase sites were present in the C-terminus of Hst4; the levels of Hst4 were determined in the *hsk1-89* in which Hsk1 is active at 25 degrees but inactive at 30 degrees centigrade. Hst4 was significantly stable in this mutant even after 4 hr of MMS treatment (*Figure 3D*). To check whether the level of Hst4 is altered in mutants where replication

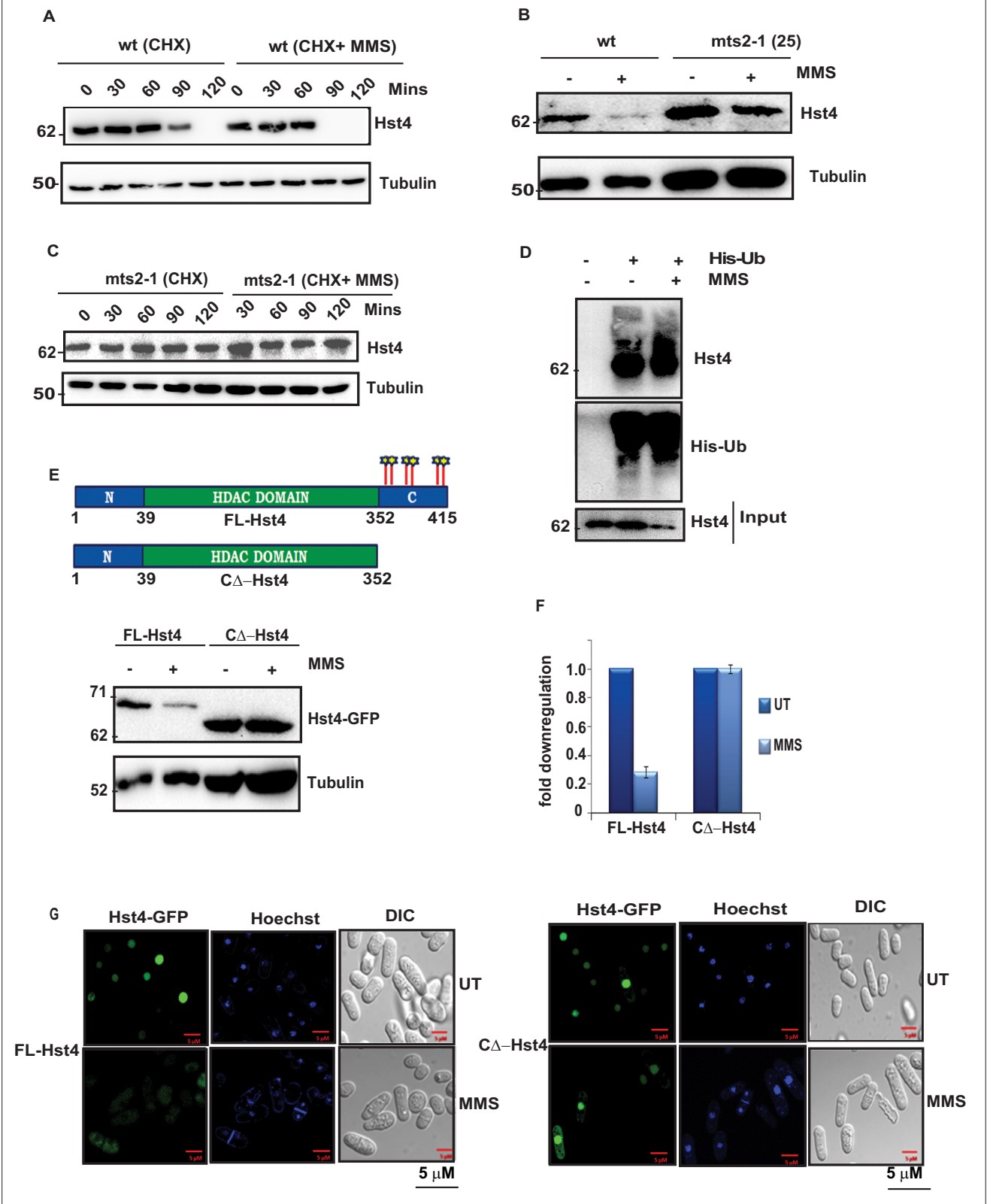

**Figure 2.** The C terminus of Hst4 is required for proteasome dependent degradation of Hst4 upon replication stress. (**A**) Half-life of Hst4 was measured by growing wt (ROP238) cells to O.D 0.5 and treating with cycloheximide (CHX) at100 μg/ml in the presence and absence of 0.015 % MMS. Cells were collected at indicated timepoints and western blot was performed. (**B**) wt strain (ROP238) containing TAP-tagged Hst4 and *mts2-1* strain (DHP37) were grown to 0.5 O.D and treated with 0.015 % MMS for 2 hr and whole cell extracts were prepared and immunoblotted with anti-PAP antibody. The

*Figure 2 continued on next page*

*Figure 2 continued*

treatment was performed at 25 °C. (**C**) *mts2-1* strain (DHP37) was grown to 0.5 O.D and cycloheximide (CHX) treatment at 100 µg/ml in the presence and absence of 0.015 % MMS was done, cells were collected at indicated timepoints and western blot was performed. The treatment was performed at non-permissive temperature (36 °C). (**D**) *mts2-1* strain expressing pREP-6XHis-Ub or empty vector were grown to OD 0.5 and shifted to 36 °C for 1 hr followed by treatment with 0.015 % MMS for 2 hr or kept untreated. Ubiquitinated proteins were pulled down using Ni-NTA beads and western blot was performed and probed with anti-PAP antibody and anti-His antibody. (**E**) The *hst4Δ* strain (ROP57) transformed with GFP-Fl-Hst4-pSGP573 and GFP-CΔ-Hst4-pSGP573 were grown to mid-log phase in EMM-Ura medium and treated with 0.015 % MMS and whole cell extracts were prepared and western blot was performed. (**F**) Quantification of data shown in (**E**). (**G**) The *hst4Δ* strain (ROP57) transformed with GFP-Fl-Hst4 and GFP-CΔ-Hst4 were grown to mid-log phase in EMM-Ura medium, treated with 0.015 % MMS for 2 hr and live cell microscopy was performed and imaged under confocal microscope. Hoechst was used to stain the DNA.

The online version of this article includes the following source data and figure supplement(s) for figure 2:

**Source data 1.** Uncropped western blot images for *Figure 2A–E*.

**Figure supplement 1.** Hst4 is post-translationally regulated and minimum region required for Hst4 degradation lies at the C-terminus.

**Figure supplement 1—source data 1.** Uncropped western blot images for *Figure 2—figure supplement 1D*.

initiation is prevented, we checked the level of Hst4 in another replication initiation protein, *mcm4* temperature-sensitive mutant (*Matsumoto et al., 2010*), we found that Hst4 is degraded in response to replication stress in this strain at non-permissive temperature indicating Hsk1 kinase specifically regulates Hst4 (*Figure 3—figure supplement 1B*). To further substantiate our results, we complemented *hsk1-89* with wild-type Hsk1 and kinase-dead Hsk1 (K129D) cloned in pSLF272 vector and found that Hst4 degradation is rescued upon wild-type Hsk1 expression; however, the kinase-dead Hsk1 is unable to do so (*Figure 3E*). The expression of Hsk1 was checked by probing the extracts with anti-HA antibody (*Figure 3—figure supplement 1C*). There is a controversy regarding the chromatin association of DDK upon replication stress. In lower eukaryotes, it has been shown that Dbf4 (SpDfp1) and Cdc7 (SpHsk1) is dissociated from the chromatin upon replication stress; however, this is not seen in mammalian cells (*Tenca et al., 2007*; *Yamada et al., 2013*). To know the status of DDK in *S. pombe* upon replication stress, we performed chromatin fractionation of untreated and MMS treated wild-type cells. Our results have shown that neither Dfp1 nor Hsk1 is dissociated from the chromatin upon replication stress; however, we observed a significant reduction of chromatin-bound Hst4 upon MMS treatment as expected due to downregulation (*Figure 3F*). Next, we also immunoprecipitated Hst4 from both wild-type and *hsk1-89* mutant strain and looked into the phosphorylation status of Hst4 upon MMS treatment. We did not find any phosphorylated band of Hst4 in *hsk1-89* cells confirming that Hst4 to be the target of Hsk1 on MMS treatment (*Figure 3G*). The next question we asked was whether Hsk1 forms a complex with Hst4. We were unable to detect Hsk1 and Hst4 interaction in vivo indicating transient interaction. We therefore performed crosslinking by 1 % formaldehyde treatment and observed that Hst4 interacts with DDK only upon treatment with 0.015 % MMS in the wild-type cells (*Figure 3H*). To check in vitro interaction, we expressed and purified recombinant his-tagged Hst4 using Ni-NTA column, coimmunoprecipitated HA tagged-Hsk1 which also contains Dfp1-myc from HU-treated fission yeast cells using anti-HA antibody and incubated the proteins together and subjected to western blot. Our results show that Hsk1-Dfp1 forms a complex with Hst4 (*Figure 3I*). Finally, we performed an in vitro kinase assay as described in materials and methods and observed that Hsk1 phosphorylates Hst4 as heat inactivation of kinase lead to loss of phosphorylation (*Figure 3J*). The immunoprecipitated Hsk1 under the same conditions was run on gel and western blot was performed (*Figure 3—figure supplement 1D*). Overall, these results suggest that Hst4 is a physiological substrate of Hsk1, which targets it for degradation via phosphorylation.

## C-terminal motif 354-363 amino acids are critical for degradation of Hst4

The consensus sequence of DDK consists of stretches of serines/threonines close to acidic amino acids. Since, the C-terminal region (352–370 aa) of Hst4 required for degradation contained serine residues S354, S355, S362, and S363, were similar to consensus sequence of Hsk1 (*Figure 4A*), we went ahead and mutated the serines to alanines in three combinations namely; S354 and S355 in one combination (S354-355SA-Hst4), S362 and S363 in the second combination (S362-363SA-Hst4) and third where all of them were mutated to alanine (4SA-Hst4) and cloned in pSLF273 vector. We have performed

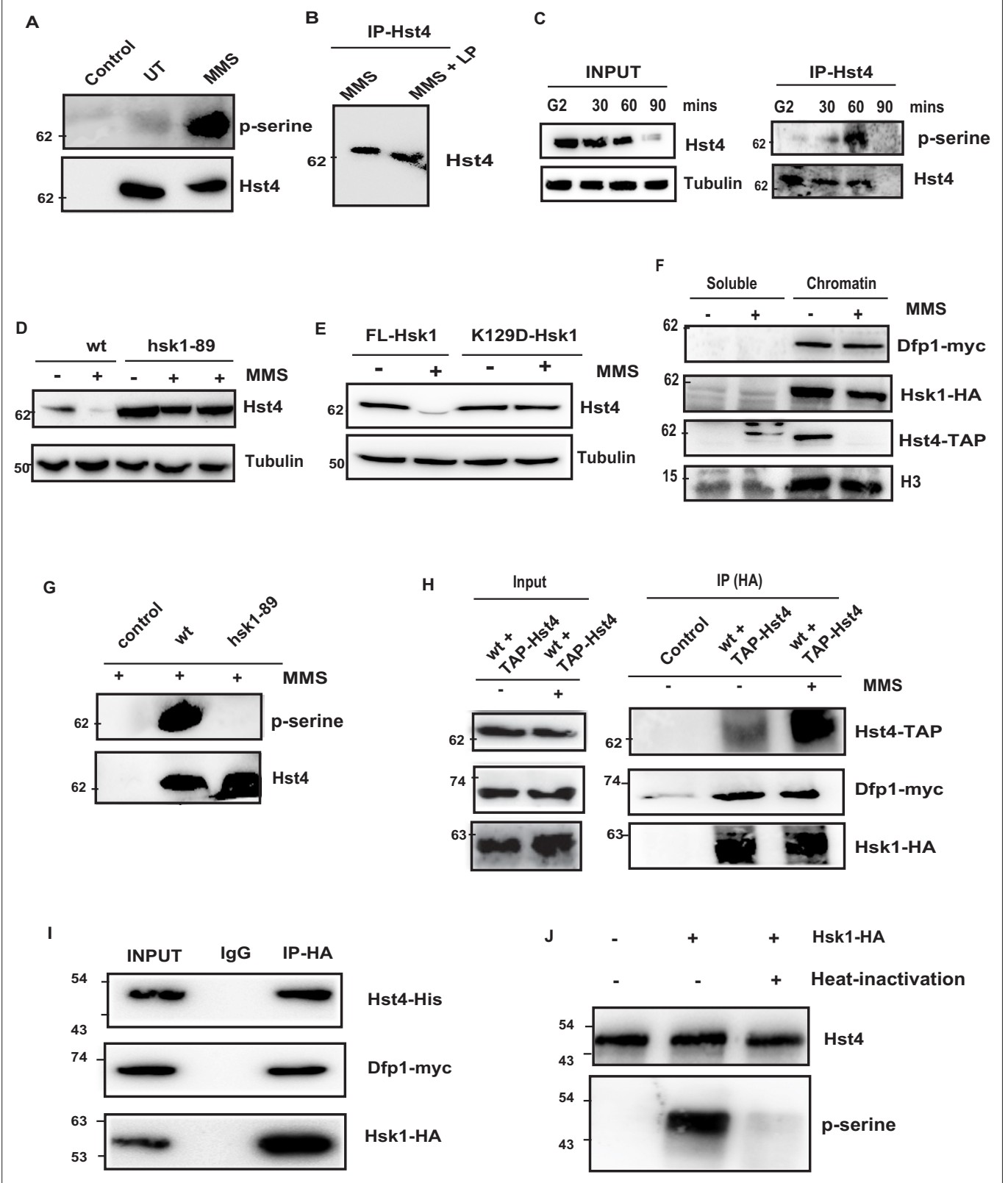

**Figure 3.** Replication-dependent regulation of Hst4 via phosphorylation by DDK/Hsk1. (**A**) wt strain (ROP238) containing TAP-tagged Hst4 was grown to 0.5 O.D., treated with 0.015 % MMS for 1 hr and samples were immunoprecipitated using IgG-sepharose beads and western blot was performed with phosphoserine antibody. Extracts from untagged strain (ROP191) were used as control. (**B**) wt strain (ROP238) were treated with 0.015 % MMS, harvested, whole cell extracts were prepared and Hst4 was immunoprecipitated using IgG sepharose beads. Half of the immunoprecipitated sample

*Figure 3 continued on next page*

*Figure 3 continued*

were kept untreated and other half was treated with lamda phosphatase for 30 min. The IP samples were boiled in 2 X SDS loading buffer and ran on 8 % SDS Polyacrylamide gel and immunoblotted. (**C**) The *cdc25-22* cells (DHP38) were first synchronized at G2 phase and then released into the S phase and cells were collected at indicated timepoints. Hst4 was immunoprecipitated using IgG-sepharose beads and western blot was performed with phosphoserine antibody. (**D**) wt (ROP238) and *hsk1-89* (DHP78) strains containing TAP-tagged Hst4 were grown to 0.5 O.D. and treated with 0.015 % MMS for indicated time points and whole cell extracts were prepared and immunoblotted with anti-PAP antibody. The *hsk1-89* strain was grown at 25 °C till O.D. 0.5 and then shifted to 30 °C for inactivation of kinase activity for 1 hr and then treated with MMS for 2 hr and 4 hr. (**E**) The complementation of *hsk1-89* strain with FL-Hsk1 (pSLF272) or kinase dead K129D-Hsk1 (pSLF272) plasmid constructs. The transformants were grown in EMM-Ura at 25 °C and treated with 0.015 % MMS for 2 hr after shifting to 30 °C, whole cell extracts were prepared and immunoblotted with anti-PAP antibody. (**F**) wt (ROP238) and DHP115 strains was grown to OD 0.5 and treated with 0.015 % MMS for 2 hr. The chromatin fractionation was performed as mentioned in the methods followed by western blot to detect indicated proteins. (**G**) wt (ROP238) and *hsk1-89* (DHP78) strains were grown and treated with 0.015 % MMS for 1 hr and immunoprecipitation of Hst4 was done. The *hsk1-89* strain was shifted to 30 °C for inactivation of kinase activity and then treated with MMS. Extracts from untagged strain (ROP191) were used as control. (**H**) The DHP115 strain containing *hsk1-HA* and *dfp1-myc* was transformed with TAP-tagged Hst4- pREP81 vector was grown to O.D. 0.5, treated with 0.015 % MMS for 1 hr and crosslinking was performed by treating with 1 % formaldehyde for 30 min. Whole cell extracts were prepared and immunoprecipitated using anti-HA antibody and western blot was performed. (**I**) Recombinant FL-Hst4 (pET28a) was expressed in BL21 cells and purified by Ni-NTA beads. The Hsk1-HA was purified from fission yeast strain containing *hsk1-HA* and *dfp1-myc* (DHP115) and bound to protein A/G beads after immunoprecipitation. Beads enriched with Hsk1 were incubated with recombinant Hst4, washed twice with lysis buffer. IgG was used as control. Western blot was performed with anti-His, anti-HA and anti-myc antibodies. (**J**) Recombinant FL-Hst4 (pET28a) was expressed in BL21 cells and purified by Ni-NTA beads. The Hsk1-HA was purified from fission yeast strain (DHP115) and in vitro kinase assay was performed as described in materials and methods.

The online version of this article includes the following source data and figure supplement(s) for figure 3:

**Source data 1.** Uncropped western blot images for *Figure 3B–J*.

**Source data 2.**

**Figure supplement 1.** Gsk3-beta and Mcm4 are not required for downregulation of Hst4.

**Figure supplement 1—source data 1.** Uncropped western blot images for *Figure 3—figure supplement 1A, B, C*.

western blot to check the stability of mutant Hst4 proteins by expressing these in *hst4Δ* strain. Our results indicate that all four residues are required for degradation (*Figure 4—figure supplement 1A*). We have made these mutations at the genomic loci, generated TAP-tagged 4SA-Hst4 expressing strain and we found similar stabilization of 4SA-Hst4 upon MMS treatment (*Figure 4B*) by western blot. We have also checked the half-life of Hst4 in wild-type and *4SA-hst4* strains and found that half-life of Hst4 is significantly increased in *4SA-hst4* indicating stabilization (*Figure 4C*). We also went ahead to perform immunoprecipitation to check the phosphorylation status of Hst4 and found that wild-type Hst4 is phosphorylated, however, the 4SA-Hst4 is not phosphorylated (*Figure 4D*). We next investigated whether Hst4 is ubiquitinated in this condition. To pull down ubiquitinated proteins, the *mts2-1* strain (DHP37) was transformed with pREP1-His6-ubiquitin or empty vector and vector with FL-Hst4 (pSLF273) or 4SA-Hst4 (pSLF273) and Ni-NTA column was used to purify the expressed ubiquitinated proteins under denaturing conditions. Our results show the loss of ubiquitinated Hst4 upon 4SA-Hst4 expression under untreated as well as treated conditions (*Figure 4—figure supplement 1B*). Immunoblotting with His antibody is shown as an expression control. Finally, we performed an in vitro phosphorylation assay using γ-P$^{32}$ ATP, purified mutated recombinant Hst4 proteins expressed in BL21 cells and Hsk1 immunoprecipitated from *S. pombe* cells and no phosphorylation was observed when 4SA-Hst4 was used as substrate (*Figure 4E*). Overall, these results confirm that Hsk1 targets Hst4 for degradation and the motif for phosphorylation and degradation lies in between serines 354–363.

## Degradation of Hst4 is required for cell survival upon DNA replication stress

Next, we wanted to understand the significance of replication stress-mediated degradation of Hst4 in cell survival. We compared the growth of CΔ-Hst4, T2Δ-Hst4 in the presence and absence of MMS. As shown in *Figure 4—figure supplement 1C*, the sensitivity and cell survival of T2-Hst4 was not altered as it was not stabilized, however, the CΔ-Hst4 mutant was severely slow growing and sensitive to MMS indicating degradation of Hst4 is required for cell survival in replication stress. The slow growth of CΔ-Hst4 could be attributed to defective downregulation of truncated Hst4 during the unperturbed

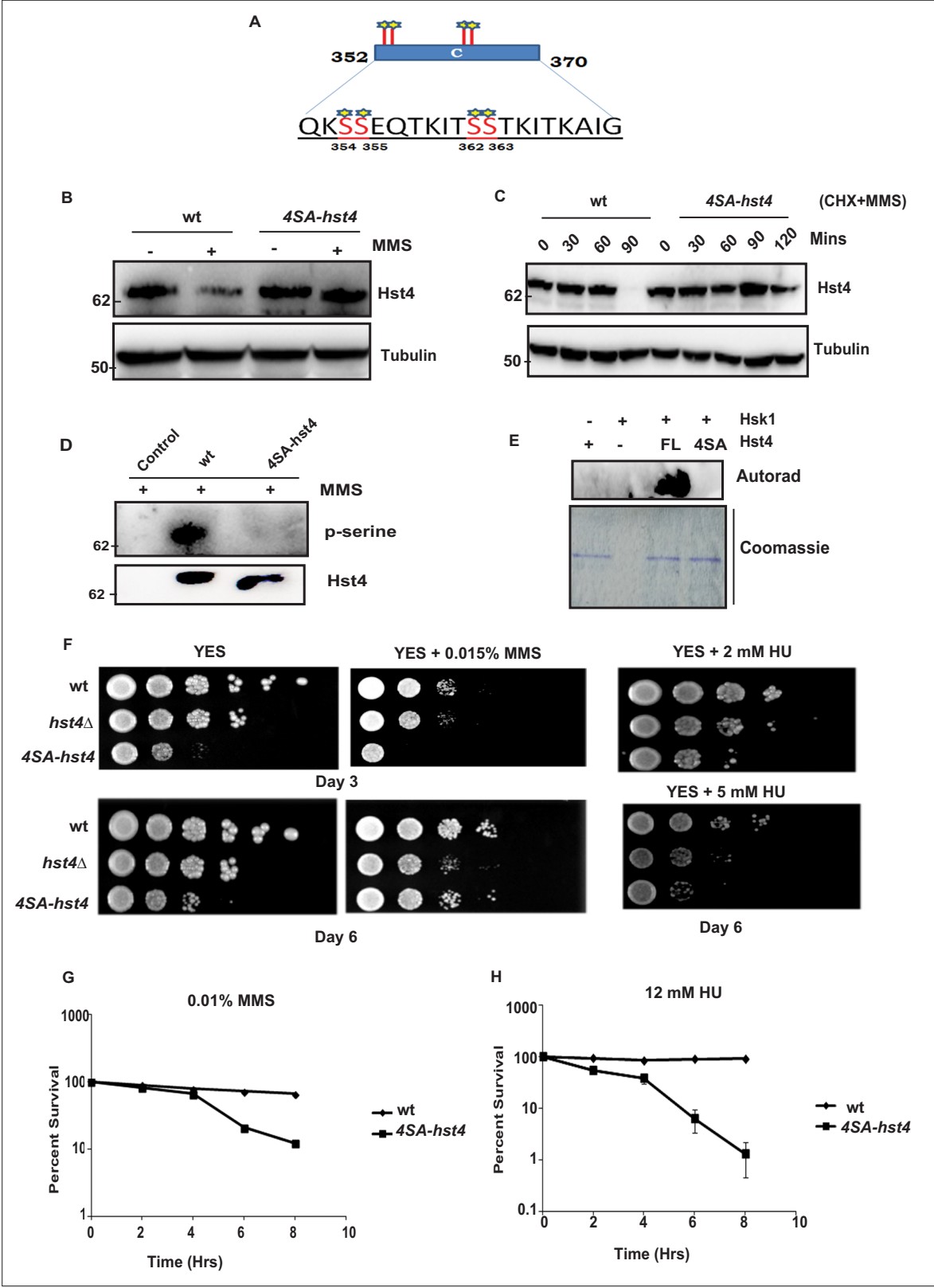

**Figure 4.** Phosphorylation of Hst4 at serine residues 354, 355, 362, and 363 is required for cell survival upon replication stress. (**A**) The C-terminal residues 352–370 required for degradation of Hst4 are shown. The serine residues marked with star, which are putative consensus DDK sites were mutated to alanines by site-directed mutagenesis. (**B**) wt strain (ROP238) and *4SA-hst4* strain (DHP146) was grown to 0.5 O.D., treated with 0.015 % MMS for 2 hr, whole cell extracts were prepared and immunoblotted with indicated antibodies. (**C**) wt (ROP238) and *4SA-hst4* (DHP146) strains were grown

*Figure 4 continued on next page*

*Figure 4 continued*

to 0.5 O.D, cycloheximide (CHX) treatment at 100 μg/ml in the presence and absence of 0.015 % MMS was done, cells were collected at indicated timepoints and western blot was performed. (**D**) wt strain (ROP238) and *4SA-hst4* strain (DHP146) was grown to 0.5 O.D., treated with 0.015 % MMS for 1 hr and samples were immunoprecipitated using IgG-sepharose beads and western blot was performed with phosphoserine antibody. TAP-tagged Hst4 was probed with anti-PAP antibody. Extracts from untagged strain (ROP191) were used as control. (**E**) Hsk1-HA was purified from DHP115 strain using anti-HA antibody. The recombinant FL-Hst4 and 4SA-Hst4 proteins were purified from BL21 cells and in vitro kinase assay was performed. The image shows autoradiogram and coomassie staining is shown for protein expression. (**F**) Spot assay showing sensitivity of non-degradable Hst4 in shown concentrations of HU and MMS containing plates. Indicated strains were grown to OD one and fivefold serial dilutions were prepared and plated onto YES, YES + HU and YES + MMS plates or EMM-Ura plates and incubated at 30 °C for 3–6 days. (**G**) (**H**) Mid-log phase cultures of wild-type and *4SA-hst4* cells were treated with 0.015 % MMS and 12 mM HU (time 0). Samples were taken at the indicated timepoints, washed free of MMS and HU, and viability was determined by colony formation on YES plates for 4 days at 30 °C. Error bars represent SEM, n = 3, and are normalized to untreated cells.

The online version of this article includes the following source data and figure supplement(s) for figure 4:

**Source data 1.** Uncropped western blot images for *Figure 4C–E*.

**Figure supplement 1.** Stability and sensitivity of Hst4 C-terminal domain deletion and serine to alanine mutants in replication stress.

**Figure supplement 1—source data 1.** Uncropped western blot images for *Figure 4—figure supplement 1A and B*.

S-phase. Further we studied the cell viability and MMS sensitivity of mutant Hst4 expression of serine to alanine point mutations in the C-terminus of Hst4 in all three combinations by expressing them in *hst4Δ* strain. To determine cell survival post replication stress, *hst4Δ* strain (ROP57) transformed with FL-Hst4, 354-355SA-Hst4, 362-363SA-Hst4, and 4SA-Hst4 (354, 355, 362, 363SA) plasmids were treated with MMS for indicated timepoints, washed off the drug, plated on fresh YES plates and counted the colonies. The result in *Figure 4—figure supplement 1D* show that *4SA-hst4* has higher recovery defect as compared to FL-Hst4 and other mutants in response to MMS treatment suggesting all four serine residues are phosphorylated to form a degron for degradation. Finally, we have checked the survival and sensitivity of endogenously expressed *4SA-hst4* mutant upon replication stress. We have performed spot assay to check the growth rate and survival upon MMS and HU treatment. As shown in *Figure 4F*, we found that cells expressing *4SA-hst4* are slow growing and sensitive to replication stress as compared to wild-type. We observed significant difference in sensitivity to MMS and HU on Day 3, however, on longer growth duration, cells adapted to MMS. We do not see such adaptation in presence of HU. The survival without stress is affected in *4SA-hst4* mutant, due to the cell cycle dependent regulation of Hst4 during S phase. To further quantitatively determine cell survival post replication stress we have treated cells with MMS and HU for indicated timepoints, cells were washed and then plated on fresh YES plates and colonies were counted. The results in *Figure 4G and H* show that *4SA-hst4* has significant growth recovery defect as compared to wild-type. We have observed that treatment with HU leads to more sensitivity and recovery defects in *4SA-hst4* mutant cells than on treatment with MMS. Overall, our results show that dynamic regulation of Hst4 is required to maintain genome stability.

## SCF(Pof3) mediated degradation of Hst4 in response to replication stress

The degradation of S phase specific substrates is targeted mainly via SCF class of E3 ubiquitin ligases which contain a Skp1/cullin core associated with the Rbx1 RING finger protein and a F box protein. The substrate specificity determinant of SCF complex is the F- box protein, which recognize sphosphorylated Ser/Thr in the target protein. . Fission yeast contains 18 ORFs encoding F-box proteins (*Hermand, 2006*). However, only a few of them have been characterized. We checked the stability of Hst4 in the absence of Pof3 (ScDia2), which is known to regulate DNA replication stress-related processes (*Roseaulin et al., 2013b*), by constructing a TAP-tagged Hst4 strain in the *pof3Δ* background. We found that Hst4 was significantly stabilized in the *pof3Δ* cells (*Figure 5A*). We have also looked at the level of Hst4 in the *pop1Δ* which is a F box protein known to affect cell ploidy (*Lehmann et al., 2004*); however, we could not find stabilization of Hst4, suggesting Pof3 is specifically required for the stabilization of Hst4 (*Figure 5—figure supplement 1A*). Further, we have examined the half-life of Hst4 in the *pof3Δ* using CHX in the presence of MMS and observed that the half life of Hst4 was significantly increased in *pof3Δ* compared to wild-type (*Figure 5B*, *Figure 5—figure supplement 1B*). Next, we confirmed the role of SCF[pof3] in the degradation of Hst4 by checking the levels of

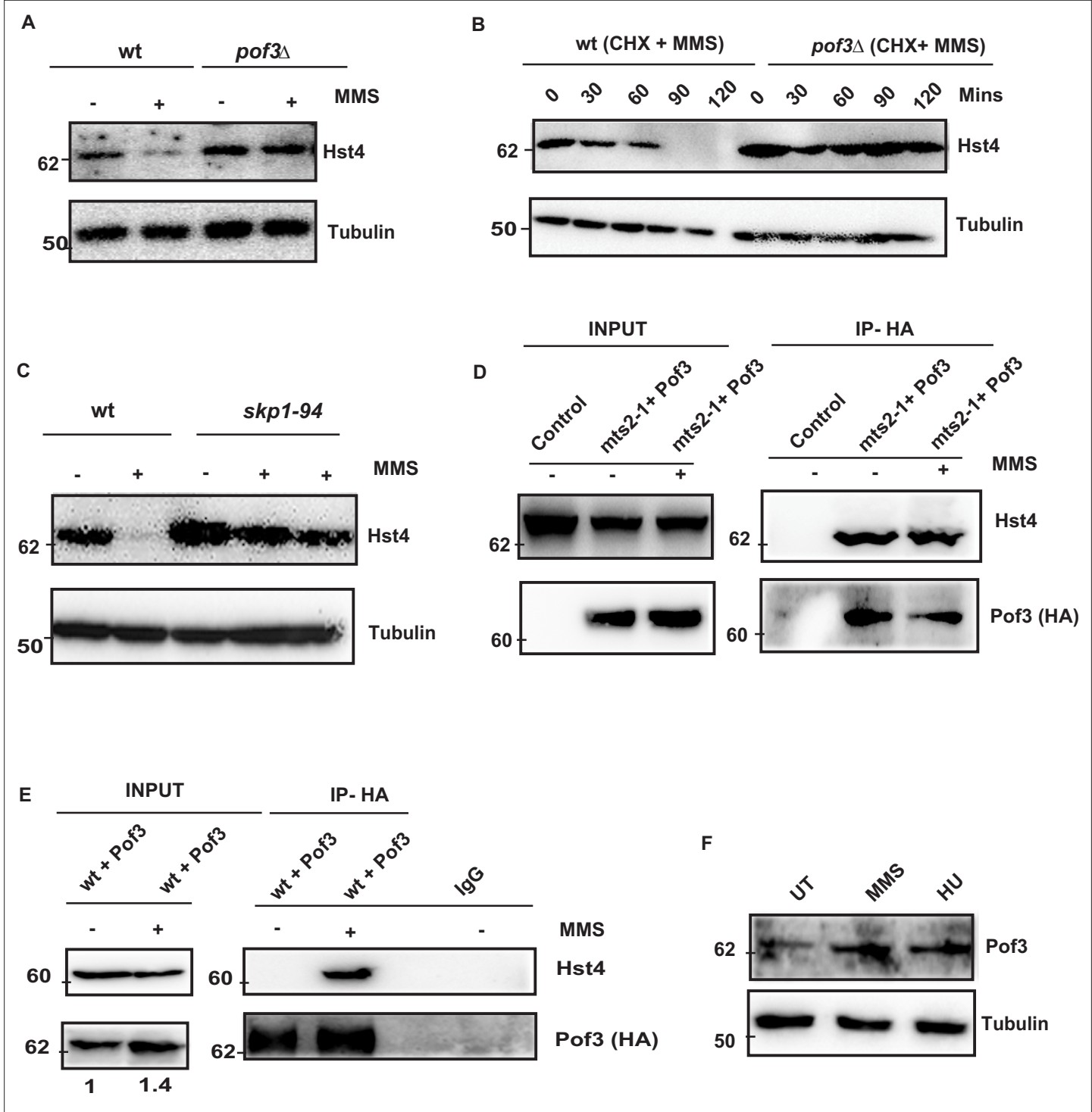

**Figure 5.** SCF[Pof3]-mediated degradation of Hst4 on replication stress. (**A**) wt strain (ROP238) and F box mutant strain *pof3Δ* (DHP35) containing TAP-tagged Hst4 were grown to 0.5 O.D., treated with 0.015 % MMS for 2 hr and whole cell extracts were prepared and immunoblotted with anti-PAP antibody. (**B**) wt strain (ROP238) and *pof3Δ* (DHP35) was grown to 0.5 O.D, treated with cycloheximide (CHX) at 100 µg/ml inpresence and absence of 0.015 % MMS, cells were collected at indicated timepoints and western blot was performed. (**C**) wt strain (ROP238) and *skp1-94* (DHP57) containing TAP-tagged Hst4 were grown to 0.5 O.D. at 25 °C, shifted to 36 °C, after 1 hr, 0.015 % MMS was added, wt cells were collected after 2 hours of treatment and *skp1-94* cells were collected after 2 and 3 hr. (**D**) HA-Pof3 in pSLF272 vector was expressed in *mts2-1* strain (DHP37). Cells were grown to midlog phase in EMM-Ura medium at 25 °C, shifted to 36 °C and after 1 hr, 0.015 % MMS was added and treated for 1 hr. Pof3 was immunoprecipitated using anti-HA antibody and western blot was performed using indicated antibodies. The extract from untransformed strain (DHP37) was used as a control. (**E**) HA-Pof3 in pSLF272 vector was expressed in wt strain (ROP238). Cells were grown to midlog phase in EMM-Ura medium and then treated with 0.015 % MMS

*Figure 5 continued on next page*

*Figure 5 continued*

for 1 hr. Pof3 was immunoprecipitated using anti-HA antibody and western blot was performed using indicated antibodies. Imunoprecipitation by IgG antibody was used as control. (**F**) Whole cell levels of Pof3 (ENY3684) upon treatment with 0.015 % MMS and 10 mM HU, respectively were checked by western blot using anti-myc antibody.

The online version of this article includes the following source data and figure supplement(s) for figure 5:

**Source data 1.** Uncropped western blot images for *Figure 5A–C and E*.

**Figure supplement 1.** Pop1 is not required for downregulation of Hst4.

**Figure supplement 1—source data 1.** Uncropped western blot images for *Figure 5—figure supplement 1C*.

Hst4 in *skp1-94* which is specifically defective in interaction with Pof3 (*Roseaulin et al., 2013a*). Our results show that Hst4 is significantly stabilized in *skp1-94* (*Figure 5C*). We did not observe stabilization of Hst4 in the *skp1-A4* strain where Skp1 interaction with Pof3 is intact (*Lehmann et al., 2004*; *Figure 5—figure supplement 1C*). Finally, we checked for interaction between Hst4 and Pof3 by expressing HA-Pof3 in *mts2-1* strain, where Hst4 is stabilized and performing coimmunoprecipitation using anti-HA antibody to pull down Pof3 after MMS treatment. We observed that Hst4 interacts with Pof3 in *mts2-1* strain without replication stress as well as upon MMS treatment (*Figure 5D*). This result suggests that Hst4 and Pof3 interaction occurs in the unperturbed S-phase cells as well. Next, we performed coimmunoprecipitation with cell extracts from MMS treated and untreated cells expressing HA-Pof3, using anti-HA antibody to check whether these proteins interact in the wild-type cells. We observed that pof3 interacts with Hst4 only in the presence of MMS in the wild-type cells upon 1 hr of treatment (*Figure 5E*). It is known that E3 ubiquitin ligases are tightly regulated by various mechanisms. Since we observed Hst4 degradation upon treatment with HU and MMS, we checked the level of Pof3 upon treatment with MMS and HU. Interestingly, we found that the Pof3 protein levels were upregulated upon MMS and HU treatment in wild-type cells (*Figure 5F*). Overall, these results further substantiate our findings on the regulation of Hst4 and establish the role of SCF^pof3 in the degradation of Hst4 in response to replication stress.

## Dynamic regulation of Hst4 is required for stable fork stalling and recovery from replication stress

To understand the physiological significance and role of DDK-mediated phosphorylation and degradation of Hst4 in cell survival upon replication stress, we investigated phenotypes of the 4SA-*hst4* strain. We performed the morphological analysis of both wild-type and 4SA-*hst4* after replication stress via Hydroxyurea-induced S phase arrest. We found a high amount of DNA fragmentation and septated cells shown by DNA staining after 6 hr of HU treatment indicating aberrant mitosis while wild-type cells were arrested, as indicated by the appearance of elongated, uninucleate cells without septa (*Figure 6A*). Since, we observed slow growth and loss of viability upon mutation of *4SA-hst4*, we studied the cell cycle progression upon the G2 block and release into the S phase in this mutant. We observed a delay in S-phase progression in this mutant (*Figure 6B*). These results indicate that degradation of Hst4 is important for maintaining genome stability during the unperturbed S phase as well as upon replication stress. Earlier observations with checkpoint proteins like *mrc1* and *rad53* have shown that lack of proper regulation leads to delayed recovery and defective restart of forks. Since Hst4 is degraded in response to replication stress; we wanted to understand the recovery kinetics after replication fork stalling. For this purpose, H2A phosphorylation status was checked as a marker of DNA damage on MMS treatment and after the removal of MMS during recovery. We found that H2A phosphorylation was reduced by 3 hr during recovery in wild-type whereas it persists in *4SA-hst4* (*Figure 6C*). These results suggest that degradation of Hst4 is required for recovery from replication stress. We have also synchronized cells in G2 phase and checked the cell cycle progression after HU-induced cell cycle arrest, to study fork recovery through flow cytometry. We observed that the wild-type cells finished S-phase and resumed cell cycle progression, however, the *4SA-hst4* showed delayed S-phase progression which further confirms the inability of this mutant to recover from HU-induced replication stress (*Figure 6—figure supplement 1A*). Further, we looked for recovery from replication stress in this strain by checking restart efficiency after replication stress through sequential incorporation of halogenated nucleotide analogs CldU and IdU and looked for colocalization.

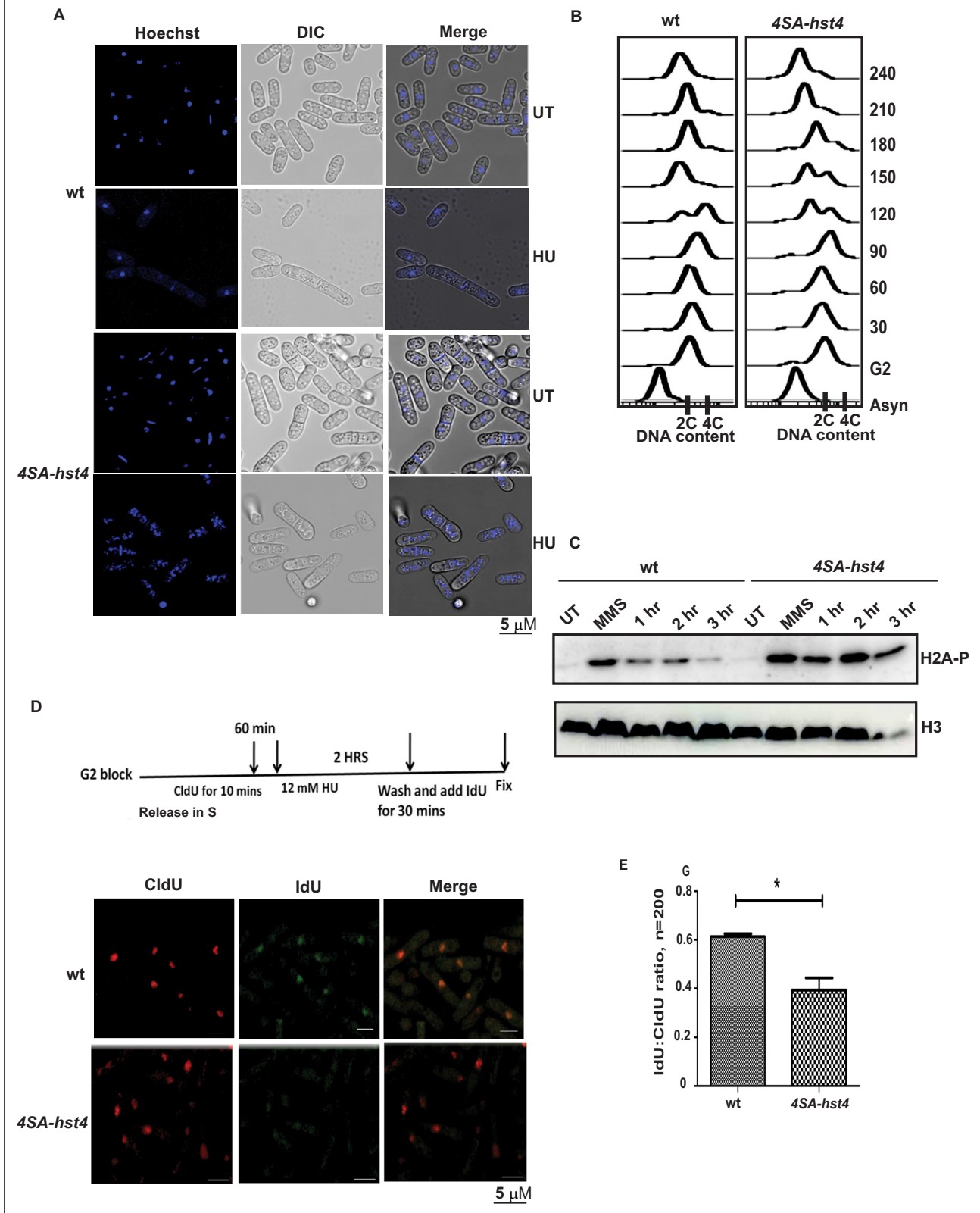

**Figure 6.** Dynamic regulation of Hst4 is required for stable fork stalling and recovery from replication stress. (**A**) Wild-type strain (ROP238) and *4SA-hst4* (DHP146) were grown to OD 0.5 and treated with 12 mM HU for 6 hr and DNA was stained with Hoechst and life cell microscopy was performed. Scale- 5 μM. (**B**) The *cdc25-22* (FY4225) and *cdc25-22* 4SA-hst4 (DHP148) grown to midlog phase in YES medium at 25 °C, were shifted to 36°C for 4 hr to synchronize at G2 phase, cells were collected every 30 min and cell cycle profile was analyzed by flow cytometry. (**C**) wt type strain (ROP238) and *4SA-*

*Figure 6 continued on next page*

*Figure 6 continued*

*hst4* (DHP146) were grown to OD 0.5 and treated with 0.03 % MMS for 2 hr, cells were washed and allowed to recover till 3 hr in fresh medium. Samples were collected at indicated timepoints and western blot was performed. (**D**) The *cdc25-22* (FY4225) and *cdc25-22* 4SA-hst4 (DHP148) strains engineered to incorporate halogenated nucleotides were grown to midlog phase in YES medium at 25 °C, were shifted to 36°C for 4 hr to synchronize at G2 phase, released into the cell cycle, 60 min post-release, CldU was added for 10 min, washed off, treated with 12 mM HU for 2 hr, washed off and allowed to recover in the presence of IdU for 30 min. Cells were fixed and immunofluorescence was performed after dual staining with CldU and IdU antibodies as mentioned in materials and methods. (**E**) The graph showing ratio of IdU to CldU indicating fork restart. A total of 100–200 cells were counted from two independent experiments and statistical analysis was performed. * indicating p < 0.05.

The online version of this article includes the following figure supplement(s) for figure 6:

**Source data 1.** Uncropped western blot images for *Figure 6C*.

**Figure supplement 1.** *4SA-hst4* is required for DNA replication fork restart post replication stress.

---

We marked replicating cells with CldU and then treated them with 12 mM HU for 2 hr and resumed replication by removing the drug and providing fresh medium with IdU. The results in *Figure 6D and E* show that the IdU:CldU ratio is significantly lower in *4SA-hst4* than that in wild-type indicating the critical need for degradation of Hst4 for fork restart. The initial replication foci detected by CldU was found to be similar in both wild-type and *4SA-hst4* implying that the early replication is not hampered in the *4SA-hst4* mutant. We have performed the restart efficiency using single analog labeling by BrdU post MMS treatment recovery in the S phase and observed that only 25 percent of cells were recovered from the stress whereas wild-type cells efficiently recovered and restarted (*Figure 6—figure supplement 1B,C*). Overall, these results suggest that degradation of Hst4 is required for proper fork stalling as well as recovery from DNA damage during the S phase.

## The degradation of Hst4 is required for stable association of the fork protection complex at the replication fork

The stability of replication fork is crucial for the maintenance of genome stability as fork collapse may result in irreversible damage to the cell, leading to death. Our data indicated that replication forks are not appropriately stalled in response to replication stress, and therefore, cells are unable to resume DNA replication efficiently. The FPC has a critical role in the stabilization of replication forks. Deficiency of FPC components Swi1 and Swi3 cause high Rad22 (Rad52) repair foci and slow S phase progression (*Noguchi et al., 2004*). The *hst4Δ* cells also exhibit similar phenotypes as shown in our earlier work (*Konada et al., 2018*). Therefore, we asked whether FPC components, *swi1*, *swi3*, and *hst4* interact genetically. We made double mutants of *swi1Δhst4Δ* and *swi3Δhst4Δ* and found that the combination is synthetic lethal as we could not recover spores (*Figure 7A*). This result indicates that these genes are functioning in parallel pathways. To check whether there is physical interaction between Hst4 and Swi1-Swi3, Swi1-FLAG and Swi3-FLAG proteins were immunoprecipitated from untreated or MMS treated fission yeast cells and incubated with recombinant Fl-Hst4. We observed that Hst4 physically interacts with Swi1 and Swi3 in vitro (*Figure 7B*). Since Hst4 is degraded upon replication stress and during S phase where these proteins have a crucial role, we hypothesized that Hst4 targets these at the chromatin via direct deacetylation. To sought that, we immunoprecipitated Swi1 and Swi3 upon MMS treatment and looked for acetylation. We did not find the acetylation of Swi1 and Swi3 (*Figure 7—figure supplement 1A*). We checked the whole cell levels of FPC in *4SA-hst4* mutant and interestingly found that Swi1 is significantly overexpressed when Hst4 was not degraded (*Figure 7C*). Mcl1, the homolog of human And-1 is also a part of FPC in higher eukaryotes. Therefore, we looked at the levels of Mcl1 and overexpression of Mcl1 was observed. The quantitation of the data is shown in *Figure 7—figure supplement 1B,C*. The replication protein Mrc1 is also part of the fork protection complex. We made a strain of *4SA-hst4* in the genetic background of *mrc1-myc* tagged strain and studied the levels of Mrc1 by western blot. We have observed that Mrc1 is also upregulated in the *4SA-hst4* strain (*Figure 7—figure supplement 1D*). We have also looked at the expression of PCNA, which is a clamp protein, crucial for DNA replication. We did not find altered expression of PCNA in *4SA-hst4* strain upon treatment with both MMS and HU which shows specific regulation of Swi1 and Mcl1 (*Figure 7—figure supplement 1E*, F). We next checked the chromatin association of FPC in the *4SA-hst4* strain and found that there is reduced level of Swi1 and Mcl1 at the chromatin when Hst4 is not degraded. (*Figure 7D*). We confirmed our results by performing

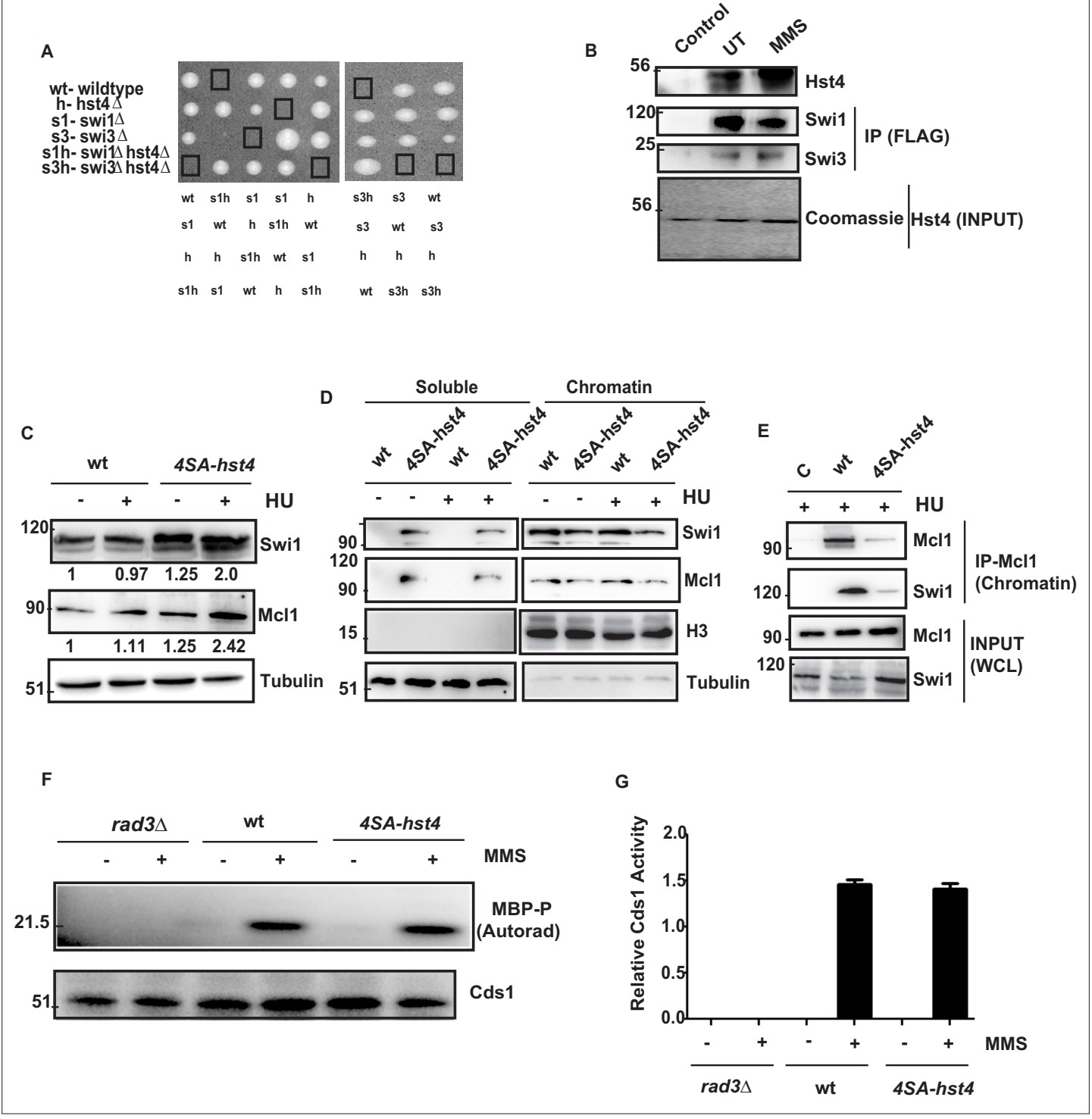

**Figure 7.** The degradation of Hst4 is required for the stable association of Fork Protection Complex at the replication fork. (**A**) Tetrads each from the *hst4Δ* strain (ROP58) crossed with the *swi1Δ* (EN3182) and *swi3Δ* (EN3366) are shown, photographed after 4 days of growth on YES plates. The genotypes were determined by replica plating on selection media are shown. (**B**) Recombinant FL-Hst4 (pET28a) was expressed in BL21 cells and purified by Ni-NTA beads. Swi1 (EN3381) or Swi3 (EN3382) FLAG-tagged strains were grown, immunoprecipitation was performed using FLAG M2 beads and incubated with recombinant Hst4 to check interaction. IgG was used as control. Western blot was performed with anti-FLAG antibody. Coomasie staining shows the expression of FL-Hst4. (**C**) wt (DHP143) and *4SA-hst4* (DHP144) with FLAG tagged Swi1 were grown to OD 0.5, treated with 12 mM HU for 2 hr, whole cell extracts were prepared and immunoblotted with indicated antibodies. (**D**) wt (DHP143) and *4SA-hst4* (DHP144) were grown to OD 0.5 and treated with 12 mM HU for 4 hr. The chromatin fractionation was performed as mentioned in the methods followed by western blot

*Figure 7 continued on next page*

*Figure 7 continued*

to detect indicated proteins. (**E**) wt (DHP143) and 4SA-Hst4 (DHP144) were grown to OD 0.5, treated with 12 mM HU for 4 hr. Chromatin fractionation was performed and chromatin associated proteins were immunoprecipitated using anti-And-1 (Mcl1) antibody. Western blot was performed using indicated antibodies. (**F**) wt (yFS988), *rad3Δ* (DHP173), and 4SA-hst4 (DHP174) containing Cds1-HA was grown till 0.5 OD and treated with 0.03 % MMS for 2 hr. Cds1 was immunoprecipitated with anti-HA antibody and in vitro Cds1 kinase activity was monitored using MBP as the substrate and γ p32 ATP was incorporated. The figure shows autoradiogram after 24 hr of exposure. (**G**) The graph showing relative kinase activity normalized to total Cds1 from two independent experiments. Error bars represent SEM.

The online version of this article includes the following source data and figure supplement(s) for figure 7:

**Source data 1.** Uncropped western blot images for *Figure 7B–D and F*.

**Figure supplement 1.** Regulation of replisome components in *4SA-hst4* mutant.

**Figure supplement 1—source data 1.** Uncropped western blot images for *Figure 7—figure supplement 1D*.

co-immunoprecipitation of chromatin enriched Mcl1, found that these proteins are less enriched in *4SA-hst4* strain (*Figure 7E*). FPC activates the intra-S phase checkpoint and stabilizes replication forks in both checkpoint dependent and independent manner. Therefore, lastly, we wanted to check the level of intra-S phase checkpoint activation in *4SA-hst4* mutant. We have grown wild-type, *rad3Δ* and *4SA-hst4* mutant strains tagged with Cds1-HA and treated with MMS and Cds1 kinase assay was performed as described in methods. We found that intra-S phase activation is normal in both wild-type and *4SA-hst4* strain (*Figure 7F and G*). We have used *rad3Δ* strain as a negative control. Overall, these results indicate that the degradation of Hst4 is important for stable association of FPC at the stalled replication fork.

## H3K56 acetylation-dependent stabilization of FPC at the chromatin regulates genomic stability

Since, we did not observe direct deacetylation of FPC by Hst4, we thought H3K56ac, which is the only known substrate of Hst4 is responsible for this reduced chromatin association. We determined the H3K56 acetylation status in *4SA-hst4* mutant and found that H3K56ac is significantly reduced in this mutant as compared to wild-type in both untreated and upon MMS treatment (*Figure 8A*). We next checked the whole cell levels of FPC in H3K56R mutants, which mimic hypoacetylation (*Haldar and Kamakaka, 2008*). We found that the whole cell levels of FPC are also upregulated in H3K56R mutant indicating the regulation of FPC could be dependent upon H3K56 acetylation (*Figure 8B*). To further confirm that the reduced chromatin association of FPC at the chromatin is dependent upon H3K56ac, we checked the chromatin association of FPC in the H3K56R mutant. We observed lesser enrichment of Swi1 and Mcl1 at the chromatin in these mutants (*Figure 8C*). We then checked the acetylation status of H3K56 under conditions where Hst4 was found to be stabilized that is in the absence of *hsk1* and *pof3*. Our results indicate that H3K56ac is drastically reduced in both *hsk1-89* and *pof3Δ* strains as a consequence of constitutive Hst4 expression in these strains (*Figure 8—figure supplement 1A, B*). Our results so far show the *S. pombe* DDK and H3K56ac-dependent regulation of FPC stability and chromatin association. It has been previously reported that the levels of FPC are highly upregulated in various types of cancer (*Bianco et al., 2019*; *Chi et al., 2017*; *Yoshida et al., 2013*). To check the conservation of regulation of FPC, we studied the levels of FPC upon knock down of Asf1a which regulates H3K56ac in the human U2OS cell line. We performed western blot to study the whole cell levels of FPC. Our results show that upon Asf1a knock down, there was a significant reduction of FPC components Tim, Tipin, and And-1 (*Figure 8D*). Since, H3K56ac is regulated by sirtuins, we checked the level of Sirtuin 2 and found that level of Sirtuin 2 was not altered upon Asf1 knock down. To confirm these results, we performed chromatin fractionation upon Asf1a knock down in U2OS cells. Our results revealed a significant decrease in the chromatin-bound Tim and And-1 consistent with the reduction in whole cell levels (*Figure 8E*). Overall, these results indicate that the regulation of FPC association to the chromatin via H3K56ac is conserved in human cells.

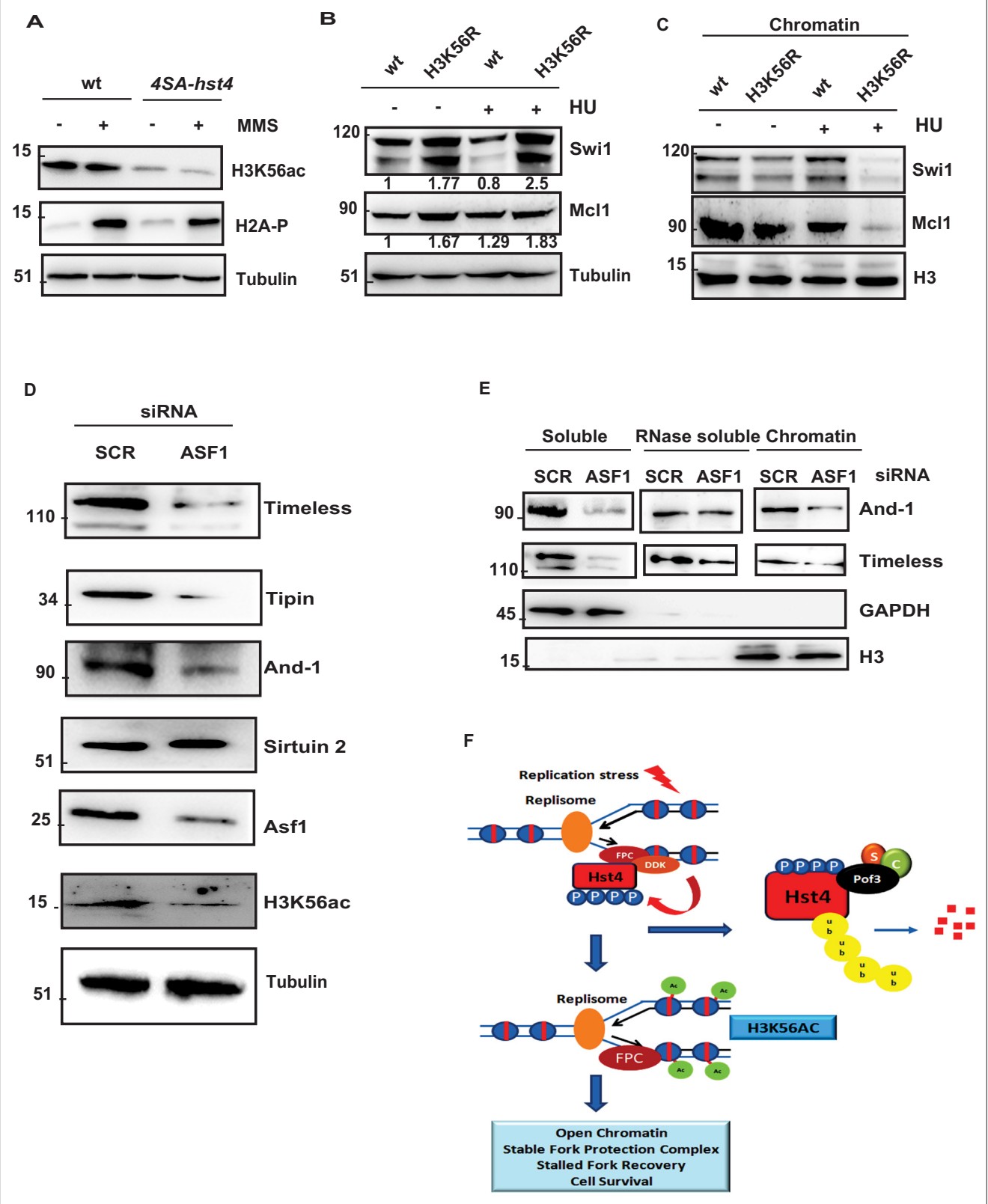

**Figure 8.** H3K56 acetylation-dependent stabilization of Fork Protection Complex at the chromatin regulates genomic stability. (**A**) wt strain (ROP238) and *4SA-hst4* strain (DHP146) were grown to 0.5 O.D., treated with 0.015 % MMS for 2 hr, whole cell extracts were prepared and immunoblotted with indicated antibodies. (**B**) wt strain and H3K56R strain (DHP169) were grown to 0.5 O.D.treated with 12 mM HU for 2 hr, whole cell extracts were prepared and immunoblotted with indicated antibodies. (**C**) wt and H3K56R strain (DHP169) was grown to 0.5 O.D. treated with 12 mM HU for 4 hr, chromatin

*Figure 8 continued on next page*

*Figure 8 continued*

fractionation followed by western blot was performed. (**D**) U2OS cells were transfected with scramble and Asf1a siRNA. At 48 hr post transfection, whole cell extracts were prepared and western blot was performed to detect indicated proteins. The expression of Asf1a and H3K56ac was detected by respective antibodies. Tubulin was checked as a loading control (**E**) To study chromatin association of FPC, U2OS cells were transfected with scramble and Asf1a siRNA. At 48 h post transfection, chromatin was extracted as described in materials and methods and western blot was performed to detect FPC components. GAPDH and H3 were Checked as controls for fractionation. (**F**) Model for the role of dynamic regulation of Hst4 in maintenance of FPC stability via H3K56ac. In S-phase or upon replication stress, DDK/Hsk1 interacts with Hst4 and phosphorylates it at serine residues at the C-terminus. The phosphorylated Hst4 is recognized and ubiquitinated by SCF[pof3] ubiquitin ligase and degraded via proteasome. Degradation of Hst4 increases H3K56ac levels leading to stable maintenance of FPC Swi1 and Mcl1 at the chromatin which helps in fork recovery and cell survival.

The online version of this article includes the following source data and figure supplement(s) for figure 8:

**Source data 1.** Uncropped western blot images for *Figure 8A–E*.

**Source data 2.**

**Figure supplement 1.** Hypoacetylation of histone H3K56 in *hsk1-89* and *pof3* mutant cells.

**Figure supplement 1—source data 1.** Uncropped western blot images for *Figure 8—figure supplement 1A and B*.

# Discussion

Several factors stabilize stalled forks against topological stress generated by pausing on the chromatinized DNA, prevent fork collapse including fork protection complex (FPC), checkpoint regulators, DDK and certain chromatin regulators. The FPC stabilizes the paused replisome via its association with MCM2 of the CMG helicase ahead of the fork which prevent fork rotation. Phosphorylation of MCM and Tof1 by DDK at the stalled fork, is required for fork stabilization in budding yeast (*Bastia et al., 2016*; *Kaplan and Bastia, 2009*). DDK, a conserved serine/threonine kinase has a significant role in the initiation of DNA replication. Studies in yeast provide substantial evidence that DDK participates in the S-phase checkpoint activation, however, it is also a target of the S phase checkpoint (*Ogi et al., 2008*; *Snaith et al., 2000*). There are contrasting reports on the role of DDK under replication stress (*Jones et al., 2021*; *Tsuji et al., 2008*; *Yamada et al., 2014*). Here, we report that DDK/Hsk1 phosphorylates sirtuin Hst4 in response to replication stress to target it for proteasomal degradation through SCF[Pof3] ubiquitin ligase to regulate replication fork recovery. The cells expressing non-degradable mutant *4SA-hst4* show reduced H3K56ac due to constitutive expression of Hst4. We propose that DDK/Hsk1 phosphorylates and targets Hst4 for degradation via SCF[Pof3] increasing H3K56ac, which facilitates stable association of FPC and its functions in stable fork stalling (*Figure 8*). We have shown that reduction in H3K56ac results in loss of FPC in human U2OS cells, indicating conservation of this mechanism, paving the path for therapeutic intervention as FPC is known to be upregulated in various cancers (*Bianco et al., 2019*; *Chi et al., 2017*; *Mazzoccoli et al., 2011*).

The genome integrity is constantly challenged by both exogenous as well as endogenous sources of replication stress. The major sensor of replication stress Rad3 (ATR in humans) gets activated upon sensing stalled replication forks and activates specific signaling pathways in response to combat it (*Cimprich and Cortez, 2008*; *Cortez, 2005*). The more recent evidence suggests S phase checkpoint-independent mechanisms also exist that contribute to the maintenance of fork stability (*Bianco et al., 2019*; *Feng et al., 2019*; *Tourrière and Pasero, 2007*; *Tourrière et al., 2005*). In *S. pombe*, it has been shown that in the absence of FPC component *swi1Δ*, cells mediate proteasome dependent degradation of replisome components to prevent fork collapse (*Roseaulin et al., 2013b*). We have earlier shown that sirtuin Hst4 gets downregulated in the S-phase of the cell cycle (*Haldar and Kamakaka, 2008*). Here, we have shown that Hst4 gets targeted for degradation upon replication fork stalling during unperturbed S phase or exogenously caused by MMS and HU. This process is dependent upon DNA replication as we failed to observe downregulation in G2 cells with damage (*Figure 1*). The DNA replication dependent phenomenon is due to the activity of DDK kinase, which is majorly active in the S phase of the cell cycle (*Matthews and Guarné, 2013*; *Ramer et al., 2013*; *Rossbach et al., 2017*). There are not many substrates of DDK known which indicate its function in replication stress response. In *S. cerevisiae*, it has been shown that DDK complex formation and activity is inhibited upon replication stress (*Kihara et al., 2000*; *Tsuji et al., 2008*; *Weinreich and Stillman, 1999*). However, more recently, it was shown that programmed fork arrest requires phosphorylation of MCM as well as Tof1 by DDK (*Bastia et al., 2016*; *Rowlands et al., 2017*).This phosphorylation is required for association

of Tof1 with the replisome. In this study, we have observed that the DDK complex is intact upon replication stress and it actively phosphorylates sirtuin Hst4 upon MMS treatment (*Figure 3*) in fission yeast, *S. pombe*. Studies in mammalian cells have also shown that DDK kinase activity and complex formation is retained and required for protecting replication forks upon replication stress (*Lee et al., 2012*). Most recent evidence has implicated the role of mammalian DDK in the replication fork restart following stalling of forks (*Jones et al., 2021*). DDK prefers S/T residues close to acidic amino acids such as aspartate and glutamate for phosphorylation (*Cho et al., 2006*; *Masai and Arai, 2002*; *Masai et al., 2006*). We observed *S. pombe* DDK/Hsk1 kinase phosphorylates serine residues S354, S355, S362, and S363 at the C-terminus of Hst4, the serines 354 and 355 are adjacent to acidic residue glutamic acid (*Figure 4*). We confirmed the stabilization of Hst4 by making a mutant *4SA-hst4*, in whichall four serines are mutated to alanines resulting in stabilization of Hst4 and loss of phosphorylation by DDK (*Figure 4*). It is to be noted that we could detect the interaction of DDK with Hst4 only upon formaldehyde crosslinking in vivo, indicating that this interaction is transient in cells. The phosphorylated residues at the C-terminus of Hst4 form a phosphodegron in order to be recognized by E3 ubiquitin ligase SCF$^{Pof3}$, where the F-box protein Pof3 recognizes phosphorylated Hst4 and degrades it via ubiquitin-dependent proteolysis (*Figures 2 and 5*). The SCF family of E3 ubiquitin ligases are very well conserved in evolution and regulate wide variety of proteins, they are also implicated in cancers (*Liu et al., 2020*; *Nakayama and Nakayama, 2006*; *Nguyen and Busino, 2020*). Pof3 regulates replication stress responses in *S. pombe* (*Katayama et al., 2002*; *Mamnun et al., 2006*). The *S. cerevisiae* homolog of Pof3, Dia2 is also a regulator of replisome integrity (*Blake et al., 2006*; *Maculins et al., 2015*; *Mimura et al., 2009*). While our work was in progress, reports on the degradation of *S. cerevisiae* Hst3, one of the functional homolog of *S. pombe* Hst4, were published (*Delgoshaie et al., 2014*; *Edenberg et al., 2014*). The degradation of Hst3 in *S. cerevisiae* is, however, carried out by a different E3 ubiquitin ligase and also the kinase phosphorylating Hst3 is different from Hsk1 indicating that the degradation mechanisms of Hst3 of *S. cerevisiae* and Hst4 of *S. pombe* are different. Our results have shown for the first time that Pof3 itself is regulated upon replication stress. We found higher protein levels of Pof3 upon both MMS and HU treatment, whereas Hst4 is targeted for degradation under the same conditions (*Figure 5F*). However, the mechanism of regulation of Pof3 upon replication stress is at present not clear and could be studied in future. In *S. pombe*, Pof3 and Hsk1 are known to regulate histone homeostasis via regulating the proteasome-dependent degradation of Ams2, the first report suggesting that DDK regulates the stability of proteins (*Takayama et al., 2010*). We believe the major function of DDK is not limited to the regulation of DNA replication, in fact, DDK is a master regulator of major DNA transactions in the cell including, DNA damage checkpoint, cohesion, homologous recombination, histone homeostasis, etc. (*Furuya et al., 2010*; *Sasi et al., 2018*; *Takayama et al., 2010*). Our findings would help highlight the positive role of DDK/Hsk1 in regulating replication stress response in chromatin context specifically in *S. pombe* and higher eukaryotes.

High amount of DNA fragmentation and recovery defects were observed when we expressed, mutant hst4, *4SA-hst4,* which was not degraded upon replication stress (*Figure 6*). The *4SA-hst4* mutant shows delayed S-phase progression as indicated by the FACS profile. Our results have also shown replication fork restart defect in *4SA-hst4* using dual labelling with BrdU analog co-localization experiments (*Figure 6*). The *4SA-hst4* mutant is not defective in initial CldU foci formation indicating that *4SA-hst4* does not have a DNA replication initiation defect. Tim, Tipin, Claspin, and And1 have been identified as members of the Replication Pausing complex (RPC), also known as FPC, which moves with the replisome and preserves replisome integrity (*Cho et al., 2013*; *Chou and Elledge, 2006*; *Errico et al., 2009*; *Errico et al., 2007*). The role in preserving the integrity of the replication fork goes beyond their function in checkpoint activation (*Errico and Costanzo, 2012*). The synthetic lethality of *hst4Δswi1Δ and hst4Δswi3Δ* observed in our study suggests that Hst4 functions in parallel with the FPC. We have earlier observed in *hst4Δ* cells, high incidence of Rad22 foci formation in the unperturbed S phase and delayed S-phase (*Konada et al., 2018*). These defects are similar to the those seen in *swi1Δ* and *swi3Δ* (*Noguchi et al., 2004*). Our current study shows that Hst4 exhibit synthetic genetic interaction and direct physical interaction with Swi1 and Swi3 (*Figure 7*). The whole cell levels of Swi1 and Mcl1 are both upregulated in *4SA-hst4* mutant. This result is consistent with our earlier study which shows that deletion of *hst4* (hyperacetylation of H3K56) leads to reduced Mcl1 levels (*Konada et al., 2018*). Interestingly, the chromatin association of these proteins is reduced upon fork stalling in *4SA-hst4* mutant, indicating phosphorylation of Hst4 is required for their chromatin

retention and fork stabilization function. Since Hst4 is a protein deacetylase, we hypothesized Swi1 and Swi3 could be its substrates; however, we did not detect direct deacetylation of these components. It is known that H3K56 acetylation is involved in the proper response to replication stress and recovery and thus completion of DNA replication (*Han et al., 2007a*; *Han et al., 2007b*). During the normal cell cycle, H3K56 is deacetylated in G2/M phase (*Maas et al., 2006*; *Masumoto et al., 2005*). This deacetylation requires Hst3 and Hst4 in *S. cerevisae*, Sir2 family deacetylases (*Maas et al., 2006*). Cells lacking Hst3 and Hst4 are sensitive to genotoxic agents that impede replication-fork progression and exhibit a high incidence of mitotic chromosome loss and replication-linked spontaneous DNA damage (*Masumoto et al., 2005*; *Wurtele et al., 2012*). Thus, the acetylation of new histones is a double-edged sword. It has also been shown that *hst3 hst4* of *S. cerevisiae* genetically interacts with Csm3 and Tof1, the homologs of Swi1 and Swi3; however, how these proteins converge in a single pathway of replication stress is not known (*Thaminy et al., 2007*). Recent Cryo-EM of FPC suggest that Tof1/Csm3 is present ahead of CMG complex and interacts with double-stranded DNA (*Baretić et al., 2020*). It will be interesting to check the interaction of FPC with nucleosomal DNA. We observed hypoacetylated H3K56 in the *4SA-hst4* as well as in *hsk1-89* and *pof3Δ* as there is constitutive expression of Hst4. Therefore; we hypothesized that the observed reduction of FPC at the chromatin and regulation of expression depends upon H3K56 acetylation. Our results in *Figure 8* have shown that the observed depletion of FPC at the chromatin in the *4SA-hst4* is indeed dependent on H3K56 acetylation as we observed similar destabilization of Swi1 and Mcl1 at the chromatin in H3K56R mutant. The whole cell levels of Swi1 and Mcl1 are also increased in H3K56R mutant indicating regulation via H3K56 acetylation. Swi1 and Swi3 have recently been shown to regulate histone H4 acetylation and physically interact with vid21 subunit of NuA4 acetyltransferase (*Noguchi et al., 2019*). Our study has further linked Swi1 and Swi3 to histone H3K56 acetylation and thereby promoting stalled replication fork stability. The *4SA-hst4* mutant has hypoacetylation of H3K56 without any exogenous replication stress indicating that forks are hypoacetylated during the unperturbed S phase, leading to cell survival defect. It is well known that the checkpoint sensor, ATR deletion leads to cell lethality indicating that ATR is active during normal S phase (*Cimprich and Cortez, 2008*; *Cortez, 2015*). The intrinsically challenged forks, therefore, need to be stabilized. It is also possible that in *4SA-hst4* mutant, chromatin is prone to collapse due to lower H3K56ac and higher expression of FPC. The evidence for this comes from the study where it has been shown that hypoacetylation of H3K56ac leads to reduced replisome components Pol €, Rfc3, and PCNA at the replication fork (*Han et al., 2007b*). Swi1 and Swi3 are known to localize in rDNA and late replicating regions and Hst4 was also observed to be localized at telomere and rDNA locus earlier (*Chang et al., 2011*). These interactions may occur in heterochromatic regions as they are prone to collapse. It is not clear yet how loss of H3K56ac leads to defective fork protection complex retention or stability at the chromatin. Further studies need to be done to know the chromatin links of FPC. A recent report has shown that cancerous cells upregulate the level of Timeless (SpSwi1) and Claspin (SpMrc1) to protect from endogenous replication stress (*Bianco et al., 2019*). We have found that the reduced levels of FPC at the chromatin, is still able to activate intra-S phase checkpoint indicating that the basal levels of FPC are enough to activate checkpoint. Similar results were obtained by *Bianco et al., 2019* wherein they show the residual levels of FPC are sufficient to activate checkpoint. It is believed that proteins of the RPC/FPC coordinate DNA polymerase and helicase activities at the stalled replication fork (*Errico et al., 2007*; *Gambus et al., 2009*; *Katou et al., 2003*; *Nedelcheva et al., 2005*). Tim-Tipin inhibits helicase activity and stimulates pol activity (*Cho et al., 2013*). During replication stress, this function becomes important to prevent excess ssDNA formation and double-strand breaks (DSBs). The non-degradable Hst4 leads to reduced FPC at the chromatin thereby the helicase and DNA polymerase uncoupling will lead to fork collapse and excess breaks indicated by high H2A-P, DNA fragmentation and defective restart in *4SA-hst4* mutant. The recent reports on chromatin modifications at the replication fork also show that DNA polymerase and helicase activities at the fork are coordinated independent of checkpoint (*Feng et al., 2019*). We believe, apart from the upstream players of checkpoint, the major functional module is dictated by these downstream players like FPC which help stabilize replication forks.

Finally, we show the conservation of regulation of FPC via the H3K56ac pathway in human U2OS cells. Upon knockdown of Asf1a, a H3-H4 chaperone which is required for H3K56ac, we found reduction in the expression of Timeless, Tipin, and And1. The chromatin levels of FPC were also drastically reduced in Asf1a knock-down conditions. This finding is important because the FPC components

Timeless and Claspin have been shown to have elevated expression in cancer cells including U2OS cells (*Bianco et al., 2019*). These proteins are controlled both transcriptionally and post-translationally (*Errico and Costanzo, 2012*; *Zhang et al., 2017*). Although Asf1a could have other functions such as nucleosome assembly, however, knock down of Asf1a here is carried out specifically, to reduce H3K56ac. The mechanism of regulation of expression of FPC via H3K56ac is currently unclear. Also the difference in the expression pattern of FPC in yeast versus mammals is intriguing. The role of H3K56ac in the regulation of transcription is well known. Interestingly, recent reports have shown that H3K56ac can have opposing effects on the transcription of genes depending upon the context of chromatin and crosstalks with other histone marks (*Cote et al., 2019*; *Topal et al., 2019*). It will be interesting to study the molecular mechanism of regulation of FPC via H3K56ac in the future. Overall, our study has shown a novel role of H3K56ac in the regulation of FPC which opens up a promising pathway to therapeutic intervention for treating cancer.

## Materials and methods

### Yeast strains, media, and growth conditions

Fission yeast strains used in this study are listed in *Table 1*. Standard techniques were used for growth, transformation and genetic manipulations (*Moreno et al., 1991*). *S. pombe* strains were grown in yeast extract plus supplements (YES) or Edinburgh minimal media (EMM) at 32 °C on the plate or in liquid media. Transformations were done using lithium acetate protocol. Ten millilitres of culture were grown to an optical density $OD_{600}$ = 1. The cells were washed with 10 ml of sterile water once followed with 5 ml of Tris-EDTA (TE) plus 0.1 M lithium acetate. Cells were resuspended in 0.1 ml of TE plus 0.1 M lithium acetate and incubated for 1 hr on a roller drum at 32 °C. 5 µl of 10 mg/ml carrier DNA (salmon sperm DNA) and 1 µg of plasmid DNA was added to 0.1 mL of cells and incubated at 32 °C for 30 min. Then, 0.7 ml of polyethylene glycol solution (40 % polyethylene glycol) was added to the cells and incubated at 32 °C for 1 hr. The cells were heat-shocked for 5 min at 42 °C, resuspended in 0.2 ml of water and plated on EMM plates supplemented without uracil. For spot dilution assays, log phase cultures were suspended at 0.4 OD600 and serially diluted fivefold onto YES (yeast extract, glucose and supplements) agar plates.

**Key resources table**

| Reagent type (species) or resource | Designation | Source or reference | Identifiers | Additional information |
|---|---|---|---|---|
| Antibody | Anti-PAP (Peroxidase Peroxidase Soluble Complex antibody produced in rabbit) | Sigma-Aldrich | Cat#P1291 RRID:AB_1079562 | WB (1:5,000) |
| Antibody | Anti-HA (rabbit polyclonal antibody) | Bethyl laboratories | Cat#A190-208A RRID:AB_67466 | WB (1:10,000) |
| Antibody | Anti-HA (rabbit polyclonal antibody) | Abcam | Cat#ab9110 RRID:AB_307019 | WB (1:5000) IP- (4 µg) |
| Antibody | Anti-H3K56ac (rabbit polyclonal antibody) | Abcam | Cat#ab71956 RRID:AB_10861799 | WB (1:2,000) |
| Antibody | Anti-H3 (rabbit polyclonal antibody) | Abcam | Cat#ab1791 RRID:AB_302613 | WB (1:5,000) |

*Continued on next page*

*Continued*

| Reagent type (species) or resource | Designation | Source or reference | Identifiers | Additional information |
|---|---|---|---|---|
| Antibody | Anti-FLAG (mouse monoclonal antibody) | Sigma-Aldrich | Cat#F1804 RRID:AB_262044 | WB (1:2,000) |
| Antibody | Anti-myc (mouse monoclonal antibody) | Sigma-Aldrich | Cat#M4439 RRID:AB_439694 | WB (1:5,000) |
| Antibody | Anti-And-1 (Mcl1) (mouse monoclonal antibody) | Biolegend | Cat#630,301 RRID:AB_2215084 | WB (1:5,000) IP- (2 µg) |
| Antibody | anti-GFP (mouse monoclonal antibody) | Cell Signalling | Cat#2,955 RRID:AB_1196614 | WB (1:2,000) IF- (1:250) |
| Antibody | Anti-His (mouse monoclonal antibody) | BD biosciences | Cat#552,565 RRID:AB_394432 | WB (1:5,000) |
| Antibody | Anti-Phosphoserine (rabbit polyclonal antibody) | Abcam | Cat#ab9332 RRID:AB_307184 | WB (1:3,00) |
| Antibody | H2A-P (rabbit polyclonal antibody) | Abcam | Cat#ab15083 RRID:AB_301630 | WB (1:5,000) |
| Antibody | Anti-BrdU (mouse monoclonal antibody) | BD biosciences | Cat#347,580 RRID:AB_400326 | IF (1:1,00) |
| Antibody | Anti-CldU (rat monoclonal antibody) | Abcam | Cat#ab6326 RRID:AB_305426 | IF (1:1,00) |
| Antibody | Anti-Timeless (rabbit monoclonal antibody) | Abcam | Cat#ab109512 RRID:AB_10863023 | WB (1:2000) |
| Antibody | Anti-Tipin | Abcam | Cat#ab229329 | WB (1:2000) |
| Antibody | Anti-Sirt2 (rabbit monoclonal antibody) | Cell Signalling | Cat#D4O5O RRID:AB_2716762 | WB (1:2500) |
| Antibody | Anti-GAPDH (rabbit monoclonal antibody) | Cell Signalling | Cat#14C10 RRID:AB_561053 | WB (1:5000) |
| Strain, strain background (*Escherichia coli*) | BL21(DE3) pLysS | Lab stock | | Chemically Competent cells |
| Cell line (*Homo-sapiens*) | U2OS (Osteosarcoma), cells | ATCC | ATCC ID: HTB-96 RRID:CVCL_0042 | Gift from Dr. Rashna Bhandari's lab |
| Transfected construct (human) | siRNA to Asf1a | Thermo Fisher Scientific | Cat#4392420 Assayid- S226044 | |
| Recombinant DNA reagent pSLF273 | | Forsburg Lab | | Cloning of FL-Hst4, T1Δ-Hst4, T2Δ-Hst4 and point mutations. |
| Recombinant DNA reagent pSLF272 | | Forsburg Lab | | Cloning of FL-Hsk1, KD-Hsk1 and Pof3 |
| Recombinant DNA reagent pET28a | | Lab stock | | Cloning of FL-Hst4 and 4SA-Hst4 |
| Recombinant DNA reagent pSGP573 | | Forsburg Lab | | Cloning of FL-Hst4 and CΔ-Hst4 |
| Recombinant DNA reagent pREP1-His6-ubiquitin | | Takashi Toda | | |
| Chemical compound, drug | Methyl methane sulphonate (MMS) | Sigma-Aldrich | Cat#129,925 | |
| Chemical compound, drug | Hydroxyurea (HU) | Sigma-Aldrich | Cat#H8627 | |
| Other | Bisbenzimide Hoechst | Sigma-Aldrich | Cat#B2261 | 10 µg/µl |
| Software, algorithm | ImageJ | ImageJ | https://imagej.net/Fiji/Downloads | Quantification of western blots |
| Software, algorithm | Graphpad Prism 6.0 | Graphpad | https://www.graphpad.com/scientific-software/prism// | Statistical analysis and graphs representation |
| Software, algorithm | FlowJo | BD Life Sciences | RRID:SCR_008520 | |

## Septation index

The strains were grown at 25 °C to log phase and synchronized in G2 by shifting the cells to 36 °C for 4 hr. Cells were then shifted to 25 °C and mitotic progression was determined by 4',6'-diamidino-2-phenylindole (DAPI) and calcofluor (50 μg/ml) staining. Three hundred cells from each time point were counted and septation index was determined by calculating the percentage of septated cells.

## Yeast spotting assay and cell survival assay

For spot dilution assays, midlog phase cultures were suspended at 1 OD600, serially diluted fivefold and spotted onto YES yeast extract, glucose and supplements or EMM agar plates. Growth was monitored at 30 °C for 3–4 days. For cell survival assay, 100–200 cells from the log phase cultures of strains were counted and plated on to the MMS and HU containing YES plates. Growth was monitored for 3–4 days.

## Protein preparation and western blotting

Total cell lysates were prepared from 10 to 50 ml culture of *S. pombe*. Cells were re-suspended in 200–400 μL of lysis buffer containing 50 mM HEPES, 500 mM NaCl, 5 mM EDTA, 0.1% NP-40, 10 % glycerol, 1 % protease inhibitor cocktail. Cells were lysed by glass beads using bead beater. Crude extracts were clarified by centrifugation and proteins were estimated through the bradford method. Samples were prepared by pre-heating proteins in SDS sample buffer (50 mMTris pH 7.5, 5 mM EDTA, 5 % SDS, 10 % glycerol, 0.5 % β-mercaptoethanol, 0.05 % bromophenol blue) and resolved by SDS-polyacrylamide gel electrophoresis (PAGE), then transferred onto PVDF membranes (Immobilon-P, Millipore); membranes were then incubated with the indicated antibodies. The bands were detected using the ECL detection reagent (Gbiosciences) with Chemidoc.

## Analysis of protein stability

Wild-type (ROP238), *mts2-1* (DHP37) and *pof3Δ* (DHP35) endogenously TAP-tagged (Hst4-TAP) cells were grown to OD 0.6 under appropriate growth conditions mentioned earlier. The HU and MMS was added at the concentration of 10 mM and 0.015 % respectively and grown for 2 hr. For checking the stability of Hst4, Cycloheximide was added at the concentration of 100 mg/ml either without or with damaging agents and cells were harvested at indicated time points. Cell extracts were prepared as described previously.

## Immunoprecipitation

For immunoprecipitation (IP) with antibody, yeast cell extracts were prepared in IP150 lysis buffer (50 mM Tris-HCl pH 7.5, 150 mM NaCl, 1% NP-40, 0.1 % SDS) according to the standard method described previously (*Moreno et al., 1991*). Soluble proteins (1–2 mg) were incubated with 1– 5 μg of antibody, immunoprecipitated proteins were run on SDS-PAGE and immunoblotting was performed. For coimmunoprecipitation of Hst4, cell extracts were prepared by lysing cells with NP-40 Phosphate lysis buffer as described earlier. 2 mg of soluble proteins were immunoprecipitated using IgG beads (GE health care). Pre-clearing of the extracts was performed for 30 mins at 4 °C. Flag IP was done after lysis using FLAG-IP lysis buffer 50 mM HEPES–KOH, pH 7.5, 500 mM NaCl, 1 mM EDTA, 1 % Triton X-100, 0.1 % sodium deoxycholate, 0.1 % SDS, 1 X protease inhibitor cocktail , 1 X phosphatase inhibitor cocktail and 80 mM β-Glycerophosphate and analysed by SDS-PAGE, followed by western blotting.

## Ubiquitination assay

Ubiquitylation analysis was performed as described previously (*Takeda and Yanagida, 2005*). 6XHis-tagged ubiquitin (His-Ub) or empty vector under the nmt1 promoter was overproduced in a wild-type strain or a proteasome defective *mts2-1* mutant strain. These cells were pre-cultured at 26 °C in the minimal EMM2 liquid media containing 2 mM thiamine, resuspended in EMM2 without thiamine, and grown overnight, and the cultures were shifted to 36⁰C for 3 hr. For the preparation of cell lysates, the cells were washed with buffer I (10 mM Tris-HCl [pH 7.5], 100 mM Na phosphate, 0.1% NP40, 10 mM Imidazole, 6 M Guanidine-HCl), lysed by vortexing with glass beads, and the lysates were cleared by centrifugation at 13,000 g for 10 min. Cell lysates were incubated for 4 hr at room temperature

with the Ni-NTA beads (Qiagen), and washed twice with buffer I and four times with buffer II (10 mM Tris-HCl [pH 7.5], 100 mM Na phosphate, 0.1% NP40, 10 mM Imidazole, 1 mM PMSF). Hst4 protein was pulled down with the IgG beads and was detected by immunoblotting with anti-PAP antibody.

## Cell cycle progression by flow cytometry

Yeast strains bearing *cdc25-22* mutation were used for synchronization. The wild-type and *cdc25-22* strain with TAP-tagged 4SA-Hst4 (Hst4-TAP) were arrested at the G2 phase by growing at 36 °C for 4 hr and then released from cell cycle arrest at 25 °C with or without MMS for 4 hr and samples were collected every 30 min. Cells were fixed with 70 % ethanol and stained with propidium iodide (PI). Flow cytometry was performed on a FACS Aria instrument using Cell Quest software. Histograms were generated using Flow JO (7.6.5) software.

## Live cell microscopy

Cells expressing FL-Hst4-GFP and CΔ-Hst4-GFP were grown to mid-log phase and treated with 0.015 % MMS. Cells were pelleted and resuspended in PBS. For staining of DNA, Hoechst solution (Sigma) was used at the concentration of 10 μg/μl. Live cell microscopy was performed on a confocal microscope. Quantification of foci was done by counting foci in three independent experiments and at least 300 cells were counted.

## Immunofluorescence

Immunofluorescence was performed as described previously (*Hodson et al., 2003*). Briefly, logarithmically growing cells (50 ml) were labeled for the indicated time at 25 °C in media containing 150 μg/ml BrdU, [Sigma], CldU, and IdU. Cells were fixed in methanol for 10 min, washed in PBS and then treated with 0.5 mg/ml Zymolyase 20T and 1 mg/mL lysing enzymes in PBS for 30 min. After washing in PBS, cells were resuspended in 1 ml of 4 N HCl and incubated for 10 min to denature the DNA. Cells were washed extensively in PBS then blocked in PBS with 10 % foetal calf serum for 1 hr. Cells were incubated overnight in BrdU antibody (BD Biosciences, 347580) at 1:100 or 1:50 in PBS with 10 % fetal calf serum and 0.05 % Tween-20. Cells were then washed in PBS and incubated with α-mouse-AlexaFluor 488 at 1:500 in PBS with 10 % foetal calf serum and 0.05 % Tween-20 for 2 h. For dual staining, CldU and IdU [Sigma] were given at the concentrations of 2 μM and 20 μM, respectively. Cells were incubated after fixing, spheroplasting and denaturation overnight in Cldu antibody (abcam, ab6326) and IdU antibody (BD Biosciences, 347580) at 1:100 and 1:50 dilution respectively in PBS with 10 % fetal calf serum and 0.05 % Tween-20. Cells were then washed in PBS and incubated with α-mouse-AlexaFluor 488 and α-rabbit-AlexaFluor 594 at 1:500 in PBS with 10 % fetal calf serum and 0.05 % Tween-20 for 2 h. Cells were washed and resuspended in PBS then put on coverslips previously treated with poly-l-lysine. DNA was detected with 4–6' diamidino-2-phenylindole (DAPI). Cells were visualized under the confocal microscope.

## Detection of phosphorylated Hst4 and phosphatase treatment

Cell lysates were prepared in phosphate buffer broken by vortexing with glass beads. The cleared lysates (2 mg of protein) were incubated with 20 ul of IgG sepharose beads. After incubation, the beads were washed four times with IPP150 buffer and incubated for 30 min at 30⁰C in the presence of 200 units of lambda phosphatase (New England Biolabs, P07535). The beads were then washed with IPP500 buffer, and samples were run on an 8 % SDS-polyacrylamide gel, followed by immunoblotting with appropriate antibodies.

## *S. pombe* chromatin fractionation

The chromatin fractionation was performed as previously described (*Kunoh and Habu, 2014*). Approximately $2.5 \times 10^8$ cells were harvested and washed in ice-cold stop buffer (150 mM NaCl, 50 mM NaF, 10 mM ethylenediaminetetraacetic acid, and 1 mM NaN₃). Cells were resuspended in 1 mL of PEMS buffer (100 mM PIPES, 1 mM EGTA, 1 mM MgCl₂, 1.2 M sorbitol, pH6.9) containing 1 mg/mL each of lysing enzyme and zymolyase 100T, followed by incubation at 37 °C for 20 min. The resulting spheroplasts were washed with 1.2 M sorbitol, then suspended in HBS-T buffer containing 25 mM MOPS, 60 mM β-glycerophosphate, 15 mM MgCl₂, 15 mM EGTA, 1 mM dithiothreitol, 0.2 mM Na₃VO₄, 1 mM phenylmethylsulfonyl fluoride, 1.2 M sorbitol, and 0.5 % Triton X-100, pH7.2 and incubated on ice for

**Table 1.** List of strains used in the study.

| Strain | Genotype | Source |
|---|---|---|
| ROP238 | h + ade6 M216 arg3-D4 his3-D1 leu1-32 ura4-D18 hst4-TAP::KanR | Lab stock |
| ROP191 | h_ ade6-210 arg3-D4 his3-D1 leu1-32 ura4-D18 | Lab stock |
| DHP38 | h + cdc25-22 ade6-216 arg3-D4 his3-D1 leu1-32 ura4-D18 hst4-TAP::KanR | This Study |
| DHP37 | h- ade6-216 arg3-D4 his3-D1 leu1-32 ura4-D18 hst4-TAP::KanR mts2-1 | This Study |
| ROP57 | h + ade6 M216 arg3-D4 his3-D1 leu1-32 ura4-D18 hst4Δ-his3+ | Lab stock |
| DHP57 | h + skp1-94 his3-D1 arg3-D4 leu1-32 ura4-D18 ade? hst4-TAP::KanR | This Study |
| DHP56b | h- skp1-A4 his3-D1 arg3-D4 leu1-32 ura4-D18 ade? hst4-TAP::KanR | This Study |
| SKP471 | h− leu1-32 ura4-D18 pof3Δ::kan | Takashi Toda |
| TP403-2B | h90 leu1-32 ura4-D18 pop1::ura+ | Takashi Toda |
| DHP35 | h- arg3-D4 his3-D1 leu1-32 ura4-D18 pof3::kanR hst4-TAP::KanR | This Study |
| DHP78 | h + hsk1-89::ura4+ his3-D1arg3-D4 leu1-32 ura4-D18 ade? hst4-TAP::KanR | This Study |
| ROP247 | h + h3.2 K56R h3. 1Δ/h4. 1Δ::his3+ h3.3Δ /h4.3Δ::arg3+ leu1-32 ura4-D18 his3-D1 arg3-D4 ade6-210 | Lab collection |
| DHP77 | h + ade6-216 arg3-D4 his3-D1 leu1-32 ura4-D18 hst4-TAP::KanR gsk3::ura4+ | This Study |
| ENY3684 | h + pof3-myc::hphMX6 pol2-3FLAG::kanMX6 leu1-32 ura4-D18 | Eishi Noguchi |
| yFS988 | h- ura4-D18 ade-? cdc10-M17 leu1-32::pYJ294(leu1 cds1-6his2HA) | Nick Rhind |
| FY4225 | h- cdc25-22 leu1::hENT-leu1+ his7-366::hsv-tk-his7+ ura4? ade6-M210 | Susan.L.Forsburg |
| FY1077 | h- hsk1HA::ura4+ ura4-D18 leu1-32 ade6-M216 | Susan Forsburg |
| FY1763 | h + leu1:dfp1 +6his3HA leu1+ dfp1-D1 ura4-D18 ade6-M216 | Susan Forsburg |
| DHP146 | h- ade6-216 arg3-D4 his3-D1 leu1-32 ura4-D18 4SA-hst4-TAP::KanR | This Study |
| DHP148 | h- cdc25-22 leu1::hENT-leu1+ his7-366::hsv-tk-his7+ ura4? ade6-M210 4SA-hst4-TAP::KanR | This Study |
| DHP168 | h- h3.1/h4. 1Δ::his3+ h3.3/h4.3Δ::arg3+ leu1-32 ura4-D18 arg3-D ade6-M210 swi1-3FLAG::KanR | This Study |
| DHP169 | h + H3.2 K56R h3.1/h4.1Δ::his3+ h3.3/h4.3Δ::arg3+ leu1-32 ura4-D18 arg3-D ade6-M210 swi1-3FLAG::KanR | This study |
| MS193 | h- leu1-32 ura4-D18 cdc21 | Hisao Masai |

*Table 1 continued on next page*

*Table 1 continued*

| Strain | Genotype | Source |
|---|---|---|
| DHP172 | h + leu1-32 ura4-D18 ade6-M210 cdc21 (mcm4) hst4-TAP::Kan | This Study |
| DHP173 | h + ura4-D18 ade-? leu1-32::pYJ294(leu1 cds1-6his2HA) rad3::ura4+ | This Study |
| DHP174 | h- ura4-D18 ade-? leu1-32::pYJ294(leu1 cds1-6his2HA) 4SA-hst4-TAP::Kan | This Study |
| EN3182 | h− swi1::Kanr leu1-32 ura4-D18 | Eishi Noguchi |
| EN3366 | h− swi3::Kanr leu1-32 ura4-D18 | Eishi Noguchi |
| EN3381 | h− swi1-3FLAG:Kanr leu1-32 ura4-D18 | Eishi Noguchi |
| EN3382 | h− swi3-3FLAG:Kanr leu1-32 ura4-D18 | Eishi Noguchi |
| DHP143 | h + leu1-32 ura4-D18 arg3-D4 ade6-M216 swi1-3FLAG::KanR | This Study |
| DHP144 | h- leu1-32 ura4-D18 arg3-D4 ade6-M216 swi1-3FLAG::KanR 4SA-hst4-TAP::KanR | This Study |
| DHP177 | h- leu1-32 ura4-D18 arg3-D4 ade6-M216 mrc1-13myc::Kan 4SA-hst4-TAP::Kan | This Study |
| DHP115a | h + hsk1HA::ura4+ ura4-D18 leu1-32 ade6-M216 dfp1-13 myc:G418r | This Study |
| DHP152 | h- leu1-32 ura4-D18 mrc1-13myc::Kan | Hisao Masai |

15 min. After centrifugation at 15,000 rpm for 15 min, the supernatant was saved as an unbound fraction. Then pellet was washed twice in HBS-T buffer, resuspended in the initial volume of HBS-T buffer, and saved as a chromatin fraction. Both fractions were subjected to immunoblotting as described above.

## DDK kinase assay

Kinase assays for Hsk1-Dfp1 were conducted as previously described (*Kakusho et al., 2008*; *Kim et al., 2008*). Twenty-five microliter reactions contained 40 mM HEPES/KOH (pH 7.6), 0.5 mM EDTA, 0.5 mM EGTA, 1 mM a-glycerophosphate, 1 mM NaF, 2 mM dithiothreitol, 8 mM MgOAc, 0.1 mM ATP, 1–2 mCi of [γ –32 P]ATP, and 5  µg of purified Hst4 protein. Reactions were incubated at 30 °C for 30 min and loaded onto SDS-PAGE. The gels were dried and subjected to autoradiography. The reactions without [γ–32P] ATP were processed for western blot and probed with phosphoserine antibody.

## Cds1 phosphorylation assay

To estimate the Cds1 kinase activity, approximately 10 OD of cells were pelleted after treatment with 0.03 % MMS. The protocol used was as described previously (*Iyer and Rhind, 2017*). Cells were lysed by bead beating in 400 µl ice-cold lysis buffer 150 mM NaCl, 50 mM Tris pH 8.0, 5 mM EDTA pH 8.0, 10 % Glycerol, 50 mM NaF, freshly added 1 mM Na3VO4 and protease inhibitor cocktail (Sigma). The lysate was cleared by centrifuging at 13,500 rpm for 10 min and combined with anti-HA Antibody (bethyl laboratories) and incubated with constant mixing at 4 °C overnight. Beads were washed twice with lysis buffer and twice with kinase buffer (5 mM HEPES pH 7.5, 37.5 mM KCl, 2.5 mM MgCl2, 1 mM DTT). The sample was then split into two portions, one was used to estimate the amount of Cds1 pulled down by western blot and the other was processed to estimate the kinase activity. For western, the Cds1 was eluted off the beads by boiling in 2 x SDS PAGE gel loading dye. Cds1 was detected by using anti-HA. For kinase assay, the beads were re-suspended in 10 µl 2 x kinase buffer, 0.5 µl 10 µCi/µl γP32-ATP, 2 µl 1 mM ATP, 5 µl 1 mg/ml myelin basic protein (Sigma) and incubated at 30 °C for 15 min. The reaction was quenched by adding SDS PAGE gel loading buffer and boiling at 98 °C for 5 min. Kinase reactions were run on a 15 % gel. The gel was dried under vacuum and exposed to a phosphoimager screen for 12 hr. The screen was scanned on Typhoon FLA-9000 and quantitated using Image J.

## Generation of Hst4 phospho-mutant strains

This FL-Hst4 gene cloned in pSLF273 plasmid was used as a template to create individual serine to alanine mutants by site-directed mutagenesis using Dpn1 method (NEB). All mutants, (please refer to *Figure 4*) were verified by nucleotide sequencing. To construct *4SA-hst4* mutant strain, the resultant plasmids were used to amplify mutant gene using a two-step PCR method combined with TAP tag and the KanMX marker. Resultant PCR fragments were then transformed into wild-type strain with G418-resistance as a selectable marker. The resultant mutant clones were confirmed by sequencing for correct targeting at Hst4 locus.

## Cell lines, siRNA treatment, and cell lysis

U2OS cells (ATCC HTB-96, verified through STR profiling) were cultured in Dulbecco's Modified Eagle's Medium (DMEM) supplemented with 10 % fetal bovine serum (FBS) and 100 U/mL penicillin and streptomycin in a humidified 5 % CO2 incubator at 37 °C. U2OS cells were transfected with both scrambled and Asf1 siRNA vector using Lipofectamine 2000 and media change was done after 6 hr. Cells were incubated for 48 h and whole cell lysates were prepared after washing with 1 X PBS and lysis in NETN lysis buffer (20 mM Tris-HCl, pH 8, 100 mM NaCl, 1 mM EDTA, 0.5 % (v/v) Nonidet P-40) containing protease and phosphatase inhibitors. The whole cell lysates were boiled for 5 min in 4 X SDS loading dye and resolved by SDS-polyacrylamide gel electrophoresis (PAGE) followed by western blotting to detect specific proteins.

## Chromatin fractionation in U2OS cells

Cellular fractionation was performed according to the previous protocol (*Drouet et al., 2005*). To obtain stringent chromatin extraction, approximately 2 × 10⁶ U2OS cells were washed and harvested in PBS followed by resuspension in buffer 1 (50 mM Hepes, pH 7.5, 150 mM NaCl, 1 mM EDTA)

containing 0.1 % Triton X-100 supplemented with protease inhibitor and phosphatase inhibitor cocktail and incubated on ice for 10 min. The suspension was centrifuged at 14,000 g for 3 min, 4 °C to collect the supernatant which contains detergent extractable fraction (Dt). The pellet was further incubated in 200 µl of extraction buffer without Triton X-100 but supplemented with 200 µg/ml RNase A (Sigma) for 30 min at 25 °C under agitation. Samples were then centrifuged at 14,000 g for 3 minutes, 4 °C to collect the RNase extractable fraction (Rn). The remaining pellet which is the RNase-resistant purified chromatin fraction (Chr) was resuspended in PBS, buffer supplemented with 1 % SDS, sonicated for 10 s and boiled for solubilization by heating at 100 °C for 5 min before western blotting.

## Statistical analysis

All graphs were made using Microsoft Excel and GraphPad Prism software. The data represents the mean value of triplicates/duplicates ± SEM. Unpaired Student t-test was used to perform the statistical analysis. The definitions for p-value is (*), p = 0.01–0.05 (significant).

## Acknowledgements

We thank Susan Forsburg, Nick Rhind, Eishi Noguchi, Hisao Masai, Kathy Gould and Takashi Toda for generously providing fission yeast strains and plasmids. The human cell line, U2OS was a gift from Rashna Bhandari at CDFD. Srithi and Bala from SEF at CDFD for the technical assistance in microscopy and FACS respectively. Sincere thanks to members of LCBE. Special mention to Mrunali Thokadiwala for preparation of SDM constructs of Hst4 in pSLF273 and Arijit Mallick for help in mammalian experiments. This study was partially supported by a grant from the Science and Engineering Research Board (SERB), Ministry of Science and Technology, India (Grant EMR/2016/003933) to DH. The research was also partially funded by grant from Centre for DNA Fingerprinting and Diagnostics (CDFD), India. https://dbtindia.gov.in/. SA is supported by UGC fellowship and CDFD core funding.

## Additional information

### Funding

| Funder | Grant reference number | Author |
|---|---|---|
| Science and Engineering Research Board | Grant EMR/2016/003933 | Devyani Haldar |
| University Grants Commission | Ref No. 23/06/2013i(EU-V) | Shalini Aricthota |

The funders had no role in study design, data collection and interpretation, or the decision to submit the work for publication.

### Author contributions

Shalini Aricthota, Conceptualization, Investigation, Methodology, Validation, Visualization, Writing – original draft; Devyani Haldar, Conceptualization, Funding acquisition, Project administration, Resources, Supervision, Writing – original draft, Writing – review and editing

### Author ORCIDs

Devyani Haldar (iD) http://orcid.org/0000-0003-1445-1374

### Decision letter and Author response

Decision letter https://doi.org/10.7554/70787.sa1
Author response https://doi.org/10.7554/70787.sa2

## Additional files

### Supplementary files

• Transparent reporting form
• Source data 1. Source data for all figures.

## Data availability

All data generated or analysed during this study are included in the manuscript and supporting files.

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
