## [Decision Letter]

**Acceptance summary:**

This manuscript brings novelties by linking chromatin features to replication fork stability. This manuscript has employed a combination of biochemistry, molecular and genetics studies to establish that, in *S. pombe*, the histone deacetylase Hst4 is targeted by DDK for proteosomal degradation during unchallenged DNA replication and in response to replication blocking agents. Preventing Hst4 degradation leads to a lower level of the histone mark H3K56Ac and impairs Swi1 and Mcl1, two components of the fork protection complex, association to chromatin, leading to stalled forks more prone to collapse. Finally, key experiments are provided to establish that reduced level of H3K56Ac in U2OS cells results in downregulation of fork protection complex.

**Decision letter after peer review:**

[Editors’ note: the authors submitted for reconsideration following the decision after peer review. What follows is the decision letter after the first round of review.]

Thank you for submitting your work entitled "DDK/Hsk1 Phosphorylates and Targets Histone Deacetylase Hst4 for Degradation to Stabilize Stalled DNA Replication Forks" for consideration by *eLife*. Your article has been reviewed by 3 peer reviewers, including Jerry L Workman as the Reviewing Editor and Reviewer #1, and the evaluation has been overseen by a Senior Editor.

Our decision has been reached after consultation between the reviewers. Based on these discussions and the individual reviews below, we regret to inform you that your work will not be considered further for publication in *eLife*.

The reviewers found the work interesting and the discovery of DDK phosphorylation of Hst4 during replication stress important. The reviewers felt that in principle the work is significant enough for consideration at *eLife*. There were however issues with the manuscript that preclude its being accepted, these include issues with the quality of some of the data, issues with apparent contradictions in some of the data and whether a strong enough case was made to conclude that Hst4 phosphorylation by DDK is replication stress activated. While the reviewers felt that with time you would be able to address many of the comments in the reviews this would clearly take longer than the three months allowed for revisions at *eLife*. Hence the manuscript has to be rejected. While this version of the manuscript is rejected, a new version of this story may be appropriate for *eLife* at some point in the future.

*Reviewer #1:*

This study in fission yeast found that the histone deacetylase hst4 is phosphorylated by the Hsk1 kinase upon replication stress. This leads to degradation of hst4 by the SCF ligase and the proteasome. H3K56 is deacetylated by hst4 and this modification is required at stalled replication forks for the resolution of the stalled forks and progression through S phase.

This is a systematic study that logically proceeds through the various steps in this pathway through a nice combination of biochemistry and genetics.

My major concern is that some of the western blots shown are of poor quality. This dampens confidence on the reproducibility of some experiments if these are the best repeats. The following blots should be repeated.

Figures 3E, 4 sup. 1A, 5 D, E and F, 7D and E.

*Reviewer #2:*

In this work, Aricthota et al. showed the fission yeast Dbf4 Dependent Kinase (DDK)/Hsk1 phosphorylates histone deacetylase Hst4 upon replication stress, which marks it for degradation by the ubiquitin ligase SCFpof3. The phosphorylation of Hst4 is required for recovery from replication stress, faulty fork restart and decreased viability. The phosphorylation of Hst4 is required for the recruitment of the fork protection complex components to the chromatin in a H3K56ac dependent manner.

The results that DDK phosphorylates Hst4 for degradation upon replication stress are interesting. However, the overall quality of the data presented is poor. Some data are not consistent and do not support the conclusion quite well. The following are my comments:

1. The authors used the MMS spot assay to examine the role of Hst4 phosphorylation in MMS sensitivity. However, the data presented are not quite convincing. The function of Hst4 phosphorylation in MMS sensitivity needs more strong evidence. For Figure 4—figure supplement 1B, the 4SA-Hst4 already grew slowly on -Ura. On -Ura+MMS plate, its slow growth cannot be simply attributed to increased sensitivity to MMS. For Figure 4—figure supplement 1B, there is no difference for FL-Hst4 and 4SA-Hst4, which argue against the role of Hst4 phosphorylation in cell function and survival. Also, there is big difference for WT and CΔ-Hst4 grown on -Ura and this difference is not augmented by MMS treatment (Figure 2—figure supplement 1F). These data point to the other function of Hst4, which compromised the main conclusion of this paper.

2. Some data presented do not support their conclusion:

For example, in Figure. 2A and B, they claimed that "Hst4 was degraded at 120 minutes in untreated wild type cells, whereas the proteolysis was accelerated upon MMS treatment (Figure. 2A and B)." However, there seems to be an opposite trend for untreated and MMS treatment. Moreover, the quantification data is not consistent with the Western blots.

Figure 5E: The Co-IP assay showed that Hst4 interacts with Pof3 and this interaction was reduced upon MMS treatment, which contradicts with the role of SCFpof3 in the degradation of Hst4 in response to replication stress.

3. There is a lot of inconsistency for data in this paper:

For example, the stability of Hst4 proteins in WT cells when treated with CHX+MMS, Figure 4C and 5C are totally different.

For Figure 2E, Hst4 was found to be ubiquitinated under normal and MMS treatment. However, in figure 2-supplement 1B, Hst4 is ubiquitinated only upon MMS treatment (Figure 2—figure supplement 1B). Please explain this discrepancy.

4. Some Co-IP data in this manuscript need to be improved. For example, in Figure 3H, they only performed in vitro co-IP by incubating recombinant purified Hst4 and purified Dfp1. What about their interaction inside cells? Moreover, the authors can examine the interaction between these proteins under different conditions, i.e. untreated and MMS treatment, HU treatment. Also, a reverse Co-IP is also recommended.

5. The phosphorylation of Hst4 needs more accurate techniques to confirm, i.e. mass spec. It is likely that not all 4 residues were phosphorylated.

6. The authors should IP Hst4 when cells were individually synchronized at G2 and S phase and then examined the phosphorylation and ubiquitination status of Hst4.

*Reviewer #3:*

The work by Haldar and colleagues describes a dissection of regulation of the histone deacetylase Hst4 by the DNA replication initiation kinase DDK, in the fission yeast model. The key findings are that Hst4 protein abundance is cell cycle regulated by ubiquitin-mediated proteolysis during S phase, triggered by the Dbf4-dependent kinase (DDK). Proteolysis of Hst4 is triggered by phosphorylation on the C-terminus by DDK, which promotes association of Hst4 with the SCF component Pof3, resulting in polyubiquitination. A phenotypic analysis of hst4 phosphorylation site mutants follows, with replication phenotypes evident, along with the expected reduction in H3K56 acetylation (the established target of Hst4). Interestingly, decreased H3K56-Ac, either by the stabilizing 4SA-hst4 mutant, or by H3K56R, correlates with increased abundance of fork protection complex proteins, but with decreased FPC proteins in the chromatin fraction.

The work presents a previously unrecognized role for DDK in controlling H3K56 acetylation via Hst4 stability. The data provide good support for Hst4 being a relevant target of DDK, and make a case for dynamic regulation of Hst4 being important. The connection between decreased H3K56-Ac and chromatin binding by the fork protection complex is interesting, but how this effect is mediated is at this point unclear. Some of the data are not convincing, particularly those that support the claim for DDK phosphorylation of Hst4 being replication stress activated. An alternative model is that down-regulation of Hst4 is important during S phase, either perturbed or unperturbed, and that the key effect of replication stress is simply to prolong S phase. There is precedent for regulated proteolysis of the related Hst3 in cerevisiae, although this is CDK-triggered and mediated by a different F-box protein (Delgoshaie 2014). Cell cycle regulation of ScHst3 and Hst4 has been described, including destabilization in MMS (Maas 2006), as has regulation of SpHst4 (Haldar 2008). The importance of dynamic regulation of ScHst3/4 was explored previously in (Maas 2006). My sense is that while the core elements of the work (DDK regulation of Hst4 abundance) are solid, the findings are not of broad enough interest or novelty (in that phospho-regulation of protein stability is common) for *eLife*.

1. Figure 1F: The DNA contents don't seem to line up properly. It appears that at 120 minutes there is a 1C peak where typically there is a 2C and 4C peak. Why in MMS is there what appears to be a delay in cell separation? How is the position of S phase defined? Accompanying the histograms with a septation index analysis would clarify the position of S.

2. Line 136: Re-word, as it sounds like the proteasome pathway ubiquitinates.

3. Line 140: What are the specifics of the bioinformatic analysis? I assume that the authors mean to say that there are putative phosphoacceptor residues in the C-term, rather than phosphorylated residues.

4. Line 149: The truncation mutant has a severe fitness defect. It is not clear from the data in supplement Figure 2F that it is MMS sensitive.

5. Line 180: S cerevisiae Dfp1 is Dbf4, not Ddc1.

6. Line 186: I don't see how it follows from the data presented that chromatin retention affords an evolutionary advantage.

7. Line 206: Again, it is not clear that the mutants confer MMS sensitivity as they have a large fitness defect unperturbed. A more quantitative assay is required here.

8. Line 219: Same comment.

9. Figure 5E: There seems to be a negative effect of MMS on the Hst4-Pof3 interaction, where increased Hst4 phosphorylation and increased Pof3 abundance would suggest that a positive effect should be evident.

10. Figure 6: Since the MMS sensitivity data are not clear, it would be of interest to look at replication phenotypes of the 4SA-hst4 mutant during unperturbed S phase.

11. Line 266: The flow cytometry data in supplement Figure 6 appear identical between wt and mutant, with very similar S phase kinetics upon release from HU. This doesn't seem consistent with defective fork restart.

12. Line 288: What is the rationale for examining hst4∆ here? The more relevant allele is 4SA-hst4.

13. Line 296: Is there statistical support for this statement?

14. Figure 7D: Is not convincing. The level of H3 in 4SA HU chromatin is decreased, casting some doubt on the lower signal for Swi1. The soluble fraction blots do not have clear bands, and the Mcl1 chromatin blot has too high background.

15. Line 300: The PCNA control experiment is in MMS rather than HU, and is not quantified, making comparisons difficult.

16. Table 1: The mating types and ade status of the strains with '?' should be determined.

17. As a general comment, bar graphs should be replaced with scatter plots showing all the data points, so that the number of replicates and their distribution is evident. Exact p-values should be stated, either in the legend or on the figures.

[Editors’ note: further revisions were suggested prior to acceptance, as described below.]

Thank you for submitting your article "DDK/Hsk1 Phosphorylates and Targets Fission Yeast Histone Deacetylase Hst4 for Degradation to Stabilize Stalled DNA Replication Forks" for consideration by *eLife*. Your article has been reviewed by 3 peer reviewers, one of whom is a member of our Board of Reviewing Editors, and the evaluation has been overseen by Jessica Tyler as the Senior Editor. The reviewers have opted to remain anonymous.

Essential revisions:

The revised manuscript by Aricthota et al., has resolved most of the concerns in the first manuscript. However, some data and images are still of poor quality. The way you assembled the figures and presented the figures needs to be improved thoroughly, including font size, italics before publication.

1. The author should define FPC in the abstract.

2. Line 81: Should be "acetylation of H3K56" instead of "acetylation of H3K56ac".

3. Figure 3J: Should examine Hsk1-HA in each reaction.

4. Figure 2G and lines 160-161. The authors provide cell images to support that GFP-Hst4 is degraded upon MMS treatment but not the C∆-Hst4. Quantification of GFP signal for each strain and condition is recommended to support reproducibility.

5. Figure 3 A and G, Figure 4D: the control lane is not described properly. According to the figure legend, all lanes should expressed Tap-Hst4 that was equally immuno-precipitated with IgG-sepharose beads. Why no Hst4 signal is detected in "control" lane. Please clarify.

6. The story suffers from some inconsistency. Hst4 is phoporylated during unchallenged S-phase but interacts with Hsk1 only upon MMS treatment. Hst4 phosphorylation is proposed to trigger Hst4 Ubiquitination by Pof3 but Ub-Hst4 is observed both in untreated and MMS-treated cells. Thus, the model on figure 8F is not supported fully by the data. The authors must establish if 4SA-Hst4 undergoes Ubiquitination to support further their model.

7. Drop tests on figure 4F and Figure 4 supplement B and C are below the quality required for publication. Because survival curves are provided to establish that 4SA-hst4 makes cells sensitive to acute exposure to MMS and HU, we would recommend the authors to remove the drop tests and to provide similar survival curves for 354-355SA-Hst4 and 362-363-Hst4 mutants.

8. Figure 5E. The control lane is also not described properly. According to the figure legend all lanes should contain HA-Pof3. For the Input panel, which band correspond to Pof3-HA (upper or lower band ?) and what is the band observed in the IP fraction ? Please clarify.

9. Figure 5D: the authors performed an HU-block and release experiment that is a classical approach to evaluate the ability of cells to recover from transient fork stalling. However, only the septum index is shown (instead of classical FACS analysis) with a poor interpretation of the data. Line 312, the authors claim that septation occurs at the end of S-phase and is an indication of completion of S-phase. This is not exact. In unchallenged cells, the formation of the septum is concomitant to S-phase. After HU treatment, cells are blocked in early S-phase, mono-nucleated and no septum. After release from HU, cells replicate and WT cells complete replication in 30-45 minutes. The occurrence of the septum after HU block is indicative of the second S-phase in WT cells, after cell division, not of the first S-phase. Thus, this paragraph must be rephrased. As it stands, the data indicate that 4SA-hst4 cells are defective at either the first S-phase and/or the first cell division. Claiming that the data indicate, "defective fork restart" (lines 314) is an over interpretation. The data presented on Figure 6E provide better indication of defective fork restart.

Figure 5E: I have two questions with this experiment. First, we guess the control should have no HA tagged-Pof3, why there is a band with the same M.W. with HA-Pof3 in control sample? Second, as Hst4 is degraded and Pof3 is increased upon exposure to MMS treatment as demonstrated in this manuscript (Figure 5F), why in the input samples, we could not see the changes?

10. Line 353: Should be Figure Supplement 1E and F.

11. Statistical analysis should be performed for some quantitative data. i.e. Figure S1A, S1B.

12. Should provide the information for antibody, i.e. H2A-P.

*Reviewer #1:*

This manuscript uncovers a novel layer of the regulation of replication fork stability, linking chromatin modifiers and the activity of components of the well-described fork protection complex. Overall, the authors provide clear and comprehensive biochemical studies to establish that Hst4 harbor 4 serine residues at its C-terminal part that are phosphorylated by DDK/Hsk1 during normal and challenged DNA replication. This phospho-degron is proposed to channel Hst4 degradation by the Ubiquitin ligase SFC-Pof3. These finding are well complemented by the genetic study of the 4SA-hst4 allele encoding a degradation resistant form of Hst4. The authors establish that the lack of Hst4 degradation leads to cellular sensitivity to replication blocking agents (MMS and HU), catastrophic mitosis upon HU treatment and defective recovery from transient fork stalling. A molecular mechanism is provided to explain those replication defects phenotypes: the lack of Hst4 degradation results in a reduced level of H3K56Ac that impairs the chromatin association of Swi1 and Mcl1, explaining the reason why stalled forks are more prone to collapse in 4SA-hst4 mutated cells. Finally, the authors provide evidences that reduced level of H3K56Ac in human cells leads to the downregulation of the fork protection complex (Tipin, And-1 and timeless). Overall, the data are convincing and provide novel insights to understand how chromatin features contributes to fork stability. Thus, this manuscript clearly brings novelties in the field. However, some parts of the manuscript require better explanations of the data obtained and/or to tone down some statements.

---

## [Author Response]

[Editors’ note: the authors resubmitted a revised version of the paper for consideration. What follows is the authors’ response to the first round of review.]

Reviewer #1:[…] My major concern is that some of the western blots shown are of poor quality. This dampens confidence on the reproducibility of some experiments if these are the best repeats. The following blots should be repeated.Figures 3E, 4 sup. 1A, 5 D, E and F, 7D and E.

We apologize for the poor quality of the blots. As suggested by the reviewer 1, we have repeated all these western blots in figures 3E, Sup. 1A, 5D, E and F, 7D and E. The current figures have been prepared with new blots to improve data quality.

Reviewer #2:[…] The results that DDK phosphorylates Hst4 for degradation upon replication stress are interesting. However, the overall quality of the data presented is poor. Some data are not consistent and do not support the conclusion quite well. The following are my comments:1. The authors used the MMS spot assay to examine the role of Hst4 phosphorylation in MMS sensitivity. However, the data presented are not quite convincing. The function of Hst4 phosphorylation in MMS sensitivity needs more strong evidence. For Figure 4—figure supplement 1B, the 4SA-Hst4 already grew slowly on -Ura. On -Ura+MMS plate, its slow growth cannot be simply attributed to increased sensitivity to MMS. For Figure 4—figure supplement 1B, there is no difference for FL-Hst4 and 4SA-Hst4, which argue against the role of Hst4 phosphorylation in cell function and survival. Also, there is big difference for WT and CΔ-Hst4 grown on -Ura and this difference is not augmented by MMS treatment (Figure 2—figure supplement 1F). These data point to the other function of Hst4, which compromised the main conclusion of this paper.

We agree with the reviewer that data shown in the Figure 4—figure supplement 1B, the growth difference was not very significant in absence and presence of MMS. To show the MMS sensitivity more convincingly, we have repeated the spot assays and checked cell growth in presence of different MMS concentrations (low MMS 0.005% and 0.015%). We have also added photographs of spot assay taken on different days to show the difference in sensitivity between FL-Hst4 and 4SA-Hst4 (Figure 4—figure supplement 1C). The difference between FL-Hst4 and 4SA-Hst4 is clear at Day 3 as well as day 6. Further, we have performed cell survival assay and studied colony formation in these strains, which is more quantitative method to check the degree of fitness difference upon replication stress (Figure 4 G, H).

In this study, we have focused on studying the function of Hst4 in replication stress response pathway where stress is caused exogenously by agents like MMS and HU. We would like to mention that replication stress can also be generated endogenously during unperturbed S phase as replication forks are known to stall during normal DNA replication. These endogenously arising stalled forks could decrease fitness of the mutant strain (4SA-Hst4 mutant) even in absence of external stress by MMS as Hst4 is not degraded during endogenous replication stress in these mutant cells. The fact that Hst4 is degraded during S phase as well as upon HU and MMS treatment shows that stalling of forks is the elicitor of degradation.

2. Some data presented do not support their conclusion:For example, in Figure. 2A and B, they claimed that "Hst4 was degraded at 120 minutes in untreated wild type cells, whereas the proteolysis was accelerated upon MMS treatment (Figure. 2A and B)." However, there seems to be an opposite trend for untreated and MMS treatment. Moreover, the quantification data is not consistent with the Western blots.

To show this result more conclusively, we have performed this experiment again. The new figure 2A and figure suppl1B shows the observed difference in the level of Hst4 at 90 minutes time point between CHX and CHX + MMS where Hst4 is significantly reduced in MMS treatment. We hope, reviewer finds this data more convincing which supports our conclusion.

Figure 5E: The Co-IP assay showed that Hst4 interacts with Pof3 and this interaction was reduced upon MMS treatment, which contradicts with the role of SCFpof3 in the degradation of Hst4 in response to replication stress.

This could be the result of the time point taken for the experiment. This experiment was done at 2 hr time point post MMS treatment where Hst4 degradation is complete and this could be the reason for decrease in interaction. We have done this experiment at 60 mins time point where we see no such reduction in the interaction between Hst4 and Pof3 (Figure 5D, E). This interaction was studied in proteasome defective mts2-1 strain where Pof3 and Hst4 interact even in absence of MMS due to the S-phase regulation of Hst4. Additionally, now we have added new data showing interaction of Hst4 and Pof3 in MMS treated conditions in wild type strain. We found that Hst4 interacts with Pof3 only upon MMS treatment in wild type cells (Figure 5E).

3. There is a lot of inconsistency for data in this paper:For example, the stability of Hst4 proteins in WT cells when treated with CHX+MMS, Figure 4C and 5C are totally different.

The stability of Hst4 is very much dependent upon cell cycle. We try our best to maintain mid log phase, however, we have observed the change in kinetics upon changes in OD between midlog to log phase cells. This could be because, the kinetics are sensitive to alteration in cell density. Quantification of this blot has been presented in Figure 5, suppl 1B.

For Figure 2E, Hst4 was found to be ubiquitinated under normal and MMS treatment. However, in figure 2-supplement 1B, Hst4 is ubiquitinated only upon MMS treatment (Figure 2—figure supplement 1B). Please explain this discrepancy.

In figure 2-supplement 1B, the immunoprecipitation was carried out with IgG beads to pulldown TAP-tagged Hst4. The experiment in Figure 2E was His-Ub pull down with Ni-NTA under denaturing condition. The difference in experimental conditions and pull down efficiency has resulted in the difference in the ubiquitinated Hst4 observed. We have removed this data from supplementary because we could not obtain the same result as in Figure 2E.

4. Some Co-IP data in this manuscript need to be improved. For example, in Figure 3H, they only performed in vitro co-IP by incubating recombinant purified Hst4 and purified Dfp1. What about their interaction inside cells? Moreover, the authors can examine the interaction between these proteins under different conditions, i.e. untreated and MMS treatment, HU treatment. Also, a reverse Co-IP is also recommended.

The interactions involving DDK are mainly transient in the cell. We have failed to observe interaction between Hst4 and Hsk1 in vivo, however, previous papers have used formaldehyde crosslinking to detect such transient interactions. We have performed formaldehyde crosslinking and confirmed in vivo interaction between Hst4 and Hsk1 (Figure 3H).

5. The phosphorylation of Hst4 needs more accurate techniques to confirm, i.e. mass spec. It is likely that not all 4 residues were phosphorylated.

We have made single site mutations of where only serine at each of these positions is mutated to alanine. We are checking the stability of mutant Hst4 currently and also confirming the phosphorylation status. Due to lockdown, we could not complete these experiments.

6. The authors should IP Hst4 when cells were individually synchronized at G2 and S phase and then examined the phosphorylation and ubiquitination status of Hst4.

We have performed immunoprecipitation in S-phase cells to detect phosphorylation of Hst4. These results are added in Figure 3C. In order to detect ubiquitination, the cdc25-22 and mts2-1 temperature sensitive strains needs to be combined to get cell cycle synchronization as well as inhibition of proteasome to detect ubiquitinated proteins. The selection of such a combined strain would be tricky and therefore, this experiment was challenging for us.

We were not successful in generating this strain for doing this experiment.

Reviewer #3:[…] The work presents a previously unrecognized role for DDK in controlling H3K56 acetylation via Hst4 stability. The data provide good support for Hst4 being a relevant target of DDK, and make a case for dynamic regulation of Hst4 being important. The connection between decreased H3K56-Ac and chromatin binding by the fork protection complex is interesting, but how this effect is mediated is at this point unclear. Some of the data are not convincing, particularly those that support the claim for DDK phosphorylation of Hst4 being replication stress activated. An alternative model is that down-regulation of Hst4 is important during S phase, either perturbed or unperturbed, and that the key effect of replication stress is simply to prolong S phase. There is precedent for regulated proteolysis of the related Hst3 in cerevisiae, although this is CDK-triggered and mediated by a different F-box protein (Delgoshaie 2014). Cell cycle regulation of ScHst3 and Hst4 has been described, including destabilization in MMS (Maas 2006), as has regulation of SpHst4 (Haldar 2008). The importance of dynamic regulation of ScHst3/4 was explored previously in (Maas 2006). My sense is that while the core elements of the work (DDK regulation of Hst4 abundance) are solid, the findings are not of broad enough interest or novelty (in that phospho-regulation of protein stability is common) for eLife.1. Figure 1F: The DNA contents don't seem to line up properly. It appears that at 120 minutes there is a 1C peak where typically there is a 2C and 4C peak. Why in MMS is there what appears to be a delay in cell separation? How is the position of S phase defined? Accompanying the histograms with a septation index analysis would clarify the position of S.

We thanks reviewer for pointing out the problem in marking the DNA content. We apologize for the mistake. We have corrected the labelling of the peaks. We have added septation index analysis data for this experiment using which S-phase was marked (Figure 1, Suppl Figure 1B).

2. Line 136: Re-word, as it sounds like the proteasome pathway ubiquitinates.

We have rewritten this line.

3. Line 140: What are the specifics of the bioinformatic analysis? I assume that the authors mean to say that there are putative phosphoacceptor residues in the C-term, rather than phosphorylated residues.

We have added the details of the bioinformatics method used in the text.

4. Line 149: The truncation mutant has a severe fitness defect. It is not clear from the data in supplement Figure 2F that it is MMS sensitive.

We have placed this data in Figure 4, Suppl 1B along with other MMS sensitivity data sets. We have added the photograph of spot assay data on day 3 and day 6. On day 3, significant MMS sensitivity was observed. The difference is less (5 fold) on day 6 (Figure 4, Suppl 1B).

5. Line 180: S cerevisiae Dfp1 is Dbf4, not Ddc1.

We are sorry for this mistake. We have corrected this mistake.

6. Line 186: I don't see how it follows from the data presented that chromatin retention affords an evolutionary advantage.

We agree with the reviewer. We have removed this line.

7. Line 206: Again, it is not clear that the mutants confer MMS sensitivity as they have a large fitness defect unperturbed. A more quantitative assay is required here.8. Line 219: Same comment.

We agree with the reviewer that data shown in the Figure 4—figure supplement 1B, the growth difference was not very significant in absence and presence of MMS. To show the MMS sensitivity more convincingly, we have repeated the spot assays and checked cell growth in presence of different MMS concentrations (low MMS 0.005% and 0.015%). We have also added photographs of spot assay taken on different days to show the difference in sensitivity between FL-Hst4 and 4SA-Hst4 (Figure 4—figure supplement 1C). The difference between FL-Hst4 and 4SA-Hst4 is clear at Day 3 as well as day 6. Further, we have performed cell survival assay and studied colony formation in these strains, which is more quantitative method to check the degree of fitness difference upon replication stress (Figure 4 G, H).

In this study, we have focused on studying the function of Hst4 in replication stress response pathway where stress is caused exogenously by agents like MMS and HU. We would like to mention that replication stress can also be generated endogenously during unperturbed S phase as replication forks are known to stall during normal DNA replication. These endogenously arising stalled forks could decrease fitness of the mutant strain (4SA-Hst4 mutant) even in absence of external stress by MMS as Hst4 is not degraded during endogenous replication stress in these mutant cells. The fact that Hst4 is degraded during S phase as well as upon HU and MMS treatment shows that stalling of forks is the elicitor of degradation.

9. Figure 5E: There seems to be a negative effect of MMS on the Hst4-Pof3 interaction, where increased Hst4 phosphorylation and increased Pof3 abundance would suggest that a positive effect should be evident.

This is the result of the time point of MMS treatment taken for the experiment. This experiment was done at 2 hr timepoint post MMS treatment where Hst4 degradation is complete and this could be the reason for decrease in interaction. Now, we have done this experiment at 60 mins timepoint where we see no such reduction in the interaction between Hst4 and Pof3.

10. Figure 6: Since the MMS sensitivity data are not clear, it would be of interest to look at replication phenotypes of the 4SA-hst4 mutant during unperturbed S phase.

We have already performed cell cycle analysis on the 4SA mutant. We have also checked DNA synthesis using incorporation of nucleotide analogs (IdU and CldU, Figure 6) which shows that both unperturbed as well as damage phenotypes are affected.

11. Line 266: The flow cytometry data in supplement Figure 6 appear identical between wt and mutant, with very similar S phase kinetics upon release from HU. This doesn't seem consistent with defective fork restart.

There is a difference in the number of cells which have progressed through the cell cycle in wt and 4SA-hst4 after HU treatment indicated by septation analysis. If we take more time points, this may become more clear. As it is clear from septation index, we have removed this data from supplementary figure.

12. Line 288: What is the rationale for examining hst4∆ here? The more relevant allele is 4SA-hst4.13. Line 296: Is there statistical support for this statement?

We have three repeats for this data. We have provided the p values.

14. Figure 7D: Is not convincing. The level of H3 in 4SA HU chromatin is decreased, casting some doubt on the lower signal for Swi1. The soluble fraction blots do not have clear bands, and the Mcl1 chromatin blot has too high background.

We have repeated this experiment and added a new improved data. Hope the reviewer will find this convincing.

15. Line 300: The PCNA control experiment is in MMS rather than HU, and is not quantified, making comparisons difficult.

We have added data showing PCNA control experiment in MMS in MMS also (Figure 7, Suppl. 1E)

16. Table 1: The mating types and ade status of the strains with '?' should be determined.

We have determined the mating type of strains.

17. As a general comment, bar graphs should be replaced with scatter plots showing all the data points, so that the number of replicates and their distribution is evident. Exact p-values should be stated, either in the legend or on the figures.

Exact p values are mentioned in the extra sheet of raw data values.

[Editors’ note: what follows is the authors’ response to the second round of review.]

Essential revisions:The revised manuscript by Aricthota et al., has resolved most of the concerns in the first manuscript. However, some data and images are still of poor quality. The way you assembled the figures and presented the figures needs to be improved thoroughly, including font size, italics before publication.1. The author should define FPC in the abstract.

We have defined FPC in the abstract as suggested by the reviewer.

2. Line 81: Should be "acetylation of H3K56" instead of "acetylation of H3K56ac".

We have incorporated this correction.

3. Figure 3J: Should examine Hsk1-HA in each reaction.

We agree with the reviewer on this point. We had the western blot data showing the level of Hsk1-HA in each reaction. We have added this data in Figure 3Supplement 1D. IgG was used as for control IP.

4. Figure 2G and lines 160-161. The authors provide cell images to support that GFP-Hst4 is degraded upon MMS treatment but not the C∆-Hst4. Quantification of GFP signal for each strain and condition is recommended to support reproducibility.

We had carried out the quantification of GFP signal for each strain and conditions but not shown the data. We have added the quantification data in supplemental figure 2G—figure supplement 1E.

5. Figure 3 A and G, Figure 4D: the control lane is not described properly. According to the figure legend, all lanes should expressed Tap-Hst4 that was equally immuno-precipitated with IgG-sepharose beads. Why no Hst4 signal is detected in "control" lane. Please clarify.

The control lane, we have loaded extract of the untagged strain fission yeast strain (ROP191), as Hst4 is not tagged, therefore, no Hst4 band was observed. We have added the details of the control in the corresponding figure legends.

6. The story suffers from some inconsistency. Hst4 is phoporylated during unchallenged S-phase but interacts with Hsk1 only upon MMS treatment. Hst4 phosphorylation is proposed to trigger Hst4 Ubiquitination by Pof3 but Ub-Hst4 is observed both in untreated and MMS-treated cells. Thus, the model on figure 8F is not supported fully by the data. The authors must establish if 4SA-Hst4 undergoes Ubiquitination to support further their model.

Similar to Pof3, it is possible that Hsk1 interacts with Hst4 in UT conditions in mts2-1 strain only. The interaction in wild type cells is highly transient and is only found post crosslinking. To clarify and further support our model, we have now performed in vivo ubiquitination assay where we observed that 4SA-Hst4 is did not get ubiquitinated and therefore, is stabilized (Figure 4—figure supplement 1B).

7. Drop tests on figure 4F and Figure 4 supplement B and C are below the quality required for publication. Because survival curves are provided to establish that 4SA-hst4 makes cells sensitive to acute exposure to MMS and HU, we would recommend the authors to remove the drop tests and to provide similar survival curves for 354-355SA-Hst4 and 362-363-Hst4 mutants.

We have not removed Figure 4F as it shows that the 4SA mutant itself is quite sensitive to MMS and HU which cannot be observed from the survival plots. We have removed the spot assays from Figure 4-Supplement 1D and we have now performed survival curves for 354-355SA-Hst4 and 362-363-Hst4 mutants as shown in the Figure 4-Supplement 1D.

8. Figure 5E. The control lane is also not described properly. According to the figure legend all lanes should contain HA-Pof3. For the Input panel, which band correspond to Pof3-HA (upper or lower band ?) and what is the band observed in the IP fraction ? Please clarify.

We apologise for the lack of clarity in the figure legend. Extract of the untransformed strain was used as control. The anti-HA antibody from Bethyl laboratories gives a non-specific band which is seen in control input lane. We have now repeated the experiment with another antibody, anti-HA antibody from Abcam, which detects only single specific band. The control IP was performed using IgG (Figure 5E).

9. Figure 5D: the authors performed an HU-block and release experiment that is a classical approach to evaluate the ability of cells to recover from transient fork stalling. However, only the septum index is shown (instead of classical FACS analysis) with a poor interpretation of the data. Line 312, the authors claim that septation occurs at the end of S-phase and is an indication of completion of S-phase. This is not exact. In unchallenged cells, the formation of the septum is concomitant to S-phase. After HU treatment, cells are blocked in early S-phase, mono-nucleated and no septum. After release from HU, cells replicate and WT cells complete replication in 30-45 minutes. The occurrence of the septum after HU block is indicative of the second S-phase in WT cells, after cell division, not of the first S-phase. Thus, this paragraph must be rephrased. As it stands, the data indicate that 4SA-hst4 cells are defective at either the first S-phase and/or the first cell division. Claiming that the data indicate, "defective fork restart" (lines 314) is an over interpretation. The data presented on Figure 6E provide better indication of defective fork restart.

We have now carried out the recovery from HU induced replication stress with G2 synchronised cells and added FACS profile (Figure 6—figure supplement 1A). A significant delay in S-phase progression post replication stress was observed in 4SA-hst4 mutant. We agree with the reviewer about the overinterpretation of septation index data and therefore, we have rephrased the text as suggested.

Figure 5E: I have two questions with this experiment. First, we guess the control should have no HA tagged-Pof3, why there is a band with the same M.W. with HA-Pof3 in control sample? Second, as Hst4 is degraded and Pof3 is increased upon exposure to MMS treatment as demonstrated in this manuscript (Figure 5F), why in the input samples, we could not see the changes?

Since the experiment was done at 1 hour post MMS treatment, the level of upregulation of Pof3 may not be so significant as compared to 2 hours treatment. The HA antibody from Bethyl laboratories gives a non-specific band which is shown in control input lane. We have now repeated the experiment with anti-HA antibody from abcam. The control lane IP was performed using IgG antibody (Figure 5E). Our data shows that Pof3 is slightly upregulated upon MMS treatment (1 hr) and interacts with Hst4. We have also quantified this data and presented the value below the blot.

10. Line 353: Should be Figure Supplement 1E and F.

We have now incorporated the suggested change. We apologise for the mistake.

11. Statistical analysis should be performed for some quantitative data. i.e. Figure S1A, S1B.

We have performed Statistical analysis as suggested.

12. Should provide the information for antibody, i.e. H2A-P.

Information about all antibodies has been provided in Table 1. Example, H2A-P antibody is from abcam (ab15083).